# Recursion in Recursion: Two-Level Nested Recursion for Length Generalization with Scalability

**Jishnu Ray Chowdhury**     **Cornelia Caragea**
Computer Science
University of Illinois Chicago
jraych2@uic.edu        cornelia@uic.edu

## Abstract

Binary Balanced Tree Recursive Neural Networks (BBT-RvNNs) enforce sequence composition according to a preset balanced binary tree structure. Thus, their non-linear recursion depth (which is the tree depth) is just $\log_2 n$ ($n$ being the sequence length). Such logarithmic scaling makes BBT-RvNNs efficient and scalable on long sequence tasks such as Long Range Arena (LRA). However, such computational efficiency comes at a cost because BBT-RvNNs cannot solve simple arithmetic tasks like ListOps. On the flip side, RvNN models (e.g., Beam Tree RvNN) that do succeed on ListOps (and other structure-sensitive tasks like formal logical inference) are generally several times more expensive (in time and space) than even Recurrent Neural Networks. In this paper, we introduce a novel framework — Recursion in Recursion (RIR) to strike a balance between the two sides - getting some of the benefits from both worlds. In RIR, we use a form of two-level nested recursion - where the outer recursion is a $k$-ary balanced tree model with another recursive model (inner recursion) implementing its cell function. For the inner recursion, we choose Beam Tree RvNNs. To adjust Beam Tree RvNNs within RIR we also propose a novel strategy of beam alignment. Overall, this entails that the total recursive depth in RIR is upper-bounded by $k \log_k n$. Our best RIR-based model is the first model that demonstrates high ($\geq 90\%$) length-generalization performance on ListOps while at the same time being scalable enough to be trainable on long sequence inputs from LRA (it can reduce the memory usage of the original Beam Tree RvNN by hundreds of times). Moreover, in terms of accuracy in the LRA language tasks, it performs competitively with Structured State Space Models (SSMs) without any special initialization - outperforming Transformers by a large margin. On the other hand, while SSMs can marginally outperform RIR on LRA, they (SSMs) fail to length-generalize on ListOps. Our code is available at: https://github.com/JRC1995/BeamRecursionFamily/.

## 1 Introduction

Non-linear Recurrent Neural Networks (RNNs) [23, 40] have the potential to enable powerful non-linear sequential dynamics that dynamically adapt with the input length. However, this ability that makes them powerful also causes several problems. First, RNNs are slow because of their non-linear sequential operations that scale linearly with the input sequence length. Second, backpropagation through arbitrarily large non-linear recurrence depth (depending on the input length) can lead to vanishing/exploding gradients [39, 3] - or can have stability issues overall [67]. Potentially, because of the combination of these factors, it was found that fast convolution-based models [99, 48, 51, 16, 57, 26, 2, 47] with fixed non-linear depth could often do just as well as an RNN. Transformers [94] are built upon this trajectory replacing bounded kernel window in convolution with an attention

37th Conference on Neural Information Processing Systems (NeurIPS 2023).

mechanism that can dynamically compute interaction weights for input tokens at any arbitrary distance.

However, despite the tremendous success of Transformers in all manner of tasks, they tend to struggle in hierarchical structure-sensitive tasks that require dynamic adaptation to input complexity and modeling of long-range dependencies [93, 82, 35, 92]. Recently, a family of models closely connected to linear RNNs [31, 73, 86, 66, 33, 30] have been emerging. Most of these models greatly outperform Transformers in long-range datasets [31, 86]. Instead of adding non-linearity in every recurrent step, they simply stack a fixed number of linear recurrences with non-linearities in between. However, in doing so these models lose the ability to adapt the non-linearity depth to input. This can make it harder to tackle tasks requiring adaptation to unseen lengths of nested non-linear operations.

In this paper, we take a step back and look again at non-linear RNNs. Non-linear RNNs, more generally, can be seen as a special case of (non-linear) Tree Recursive Neural Networks (Tree-RvNNs) [76, 87, 89, 91]. Tree-RvNNs can process sequences in an arbitrary tree-structured order. Non-linear RNNs are a special case of Tree-RvNNs following a chain-structured tree. However, such a chain-structured tree is an extreme case - requiring the maximum tree traversal height. On the other side of the extreme are balanced tree models. For instance, the height of a $k$-ary balanced tree with $n$ terminal nodes (the input length for our case) would be $log_k(n)$. Balanced Tree-RvNNs [68, 84] can then logarithmically scale with the sequence length ($n$). To an extent, this can greatly mitigate the original issues of non-linear RNNs while still preserving some degree of ability to dynamically adapt with input length. Fascinatingly, we find that a simple model like a Balanced Binary Tree RvNN (BBT-RvNN) with a modern recursive cell [82, 13] can come close to the performance of the recent strong linear RNNs or Structured State Space Models (SSMs) without much tuning or any specialized initialization on Long Range Arena (LRA) text datasets [92] - thereby, greatly outperforming Transformers in general.

Nevertheless, while BBT-RvNNs can be quite fast and scalable, they are incapable of excelling (or even passing) simple structure-sensitive tasks like ListOps[1] [70] or formal logical inference [6] in a length-generalizable manner. Transformers were already shown to underperform in these tasks [82, 93], and we find that S4D [30] (a representative of linear RNN/SSM model family) and MEGA [60] (a hybrid of Transformer and SSM) also struggle in these tasks. Looking back at Tree-RvNNs, there are several models [82, 13, 77] that dynamically learn the tree structure from the input (instead of using some enforced heuristics) and determine the order of operations accordingly. They have been able to perform well in structure-sensitive tasks. Thus, BBT-RvNNs enjoy high speed and computational efficiency but at the cost of systematic issues in modeling task structures as revealed in ListOps or logical inference tasks, whereas in sharp contrast, the more powerful RvNNs [82, 13, 77] enjoy their strong length-generalizing accuracy in ListOps, logical inference, and reasonable performance in natural language domain but at the cost of high computational resource and inability to effectively train on datasets like LRA.

To address the above challenges, we propose a new framework called Recursion in Recursion (RIR) in which we have a nested level of recursion. We use a balanced $k$-ary Tree-ordered recursion in the outer loop and we use a strong RvNN (particularly we use a Beam Tree RvNN-based model [77]) in the inner loop to process the $k$ arguments sent in from the outer loop. The inner RvNN can scale linearly with input size but now that size is bounded by a fixed constant $k$. Thus, the total non-linear depth will be bounded by a linear factor of $k \log_k(n)$ (the total outer loop being $\log_k(n)$ and the inner loop being $k$). RIR makes Beam Tree RvNN (BT-RvNN) much faster and trainable on LRA but at the cost of some accuracy loss. On the flip side, RIR can be seen as a slower version of more basic balanced tree models but at the gain of length-generalizability in ListOps and general accuracy gain in several other datasets. In addition, we identify an issue related to beam alignment in incorporating BT-RvNN within RIR and propose a strategy to circumnavigate it.

## 2 Preliminaries

**Tree Recursive Neural Networks (Tree-RvNN)**: Here we focus on constituency Tree-RvNNs [89, 91] that build up higher-order constituent representations of a given input in a bottom-up manner. Particularly, given an input sequence of vectors, Tree-RvNNs treat them as terminal nodes of a latent

---

[1]An example sample of ListOps - Input : [ $SM$ [$SM$ [$SM$ [$MAX$ 5 6 ] 2 ] 0] 5 0 8 6 ]; Output : 7. $SM$ is modulo-10 summation.

tree to be built up. The non-terminal nodes of the tree are filled sequentially by composing their immediate child nodes using some recursive cell function - $R : \mathbb{R}^d \times \mathbb{R}^d \to \mathbb{R}^d$ (we consider binary trees here so $R$ will always have only 2 children as arguments). Given some input sequence of vectors $7 + 9 \times 5 - 2$ (assume each of the symbols correspond to some vector $\in \mathbb{R}^d$), an example of Tree-RvNN-like processing can be expressed as:

$$R(R(R(7, +), R(R(9, \times), 5)), R(-, 2)) \tag{1}$$

Let us say $R$ is considered as a single non-linear neural layer. In that case, the total non-linear depth of this processing can be interpreted as the maximum nested $R$ operations which is 4 in Eqn. 1. RNNs can be now seen as a special case where the order of operation is always like this:

$$R(R(R(R(R(R(R(h_0, 7), +), 9), \times), 5), -), 2) \tag{2}$$

Here, $h_0$ is the initial hidden state for RNNs. As we can see, because of their chain-like structure RNNs will have the maximum possible recursion depth within the Tree-RvNN framework. In this case, the depth is 7. Another special case is BBT-RvNN (Balanced Binary Tree RvNN) which follows an enforced binary-balanced tree-based order of operations and can be expressed as:

$$R(R(R(7, +), R(9, \times)), R(R(5, -), 2)) \tag{3}$$

Here the maximum non-linearity depth is only 3 (i.e., $\sim \log_2(n)$; where $n = 7$). With longer sequences the differences between RNNs and BBT-RvNNs will grow bigger because BBT-RvNNs can scale logarithmically to the input length.

**Greedy Search Tree RvNN:** If our Tree-RvNN does not employ any fixed heuristic tree structure (as in RNNs or BBT-RvNNs) and if we are not relying on an external system or user to provide tree structures, we have to decide on some policy to automatically induce the latent tree structure from raw sequences. Greedy search Tree RvNNs provide us with one way of doing that. The basic idea of Greedy Search Tree RvNN can be found in multiple works [12, 88] and is related to easy-first parsing [28]. A general framework for greedy search parsing is as follows. Assume we are at some recursion step $t$. Let the input sequence of node representations in that step be $(h_1^t, h_2^t, \ldots, h_K^t)$ (where $h_i^t \in \mathbb{R}^d$). Assume we have the recursive binary cell $R$ as before and a neural $scorer$ function of the form $scorer : \mathbb{R}^{2 \times d_s} \to \mathbb{R}$ (where $d_s < d$; the role of $d_s$ will be clarified). Next, we consider all possible node pairs that can be built into a parent node. Note that for simplicity we maintain a projective tree constraint as is often standard [12, 38, 13] - so only contiguous node pairs $(h_i^t, h_{i+1}^t)$ are considered as candidates for being composed into a higher-level parent node (whose children will be those nodes). The $scorer$ assigns a scalar score to every candidate node pair as: $e_i = scorer(h_i^t[0 : d_s], h_{i+1}^t[0 : d_s])$. The slicing above is done for efficiency reasons (we set $d_s = 64$ and generally keep it smaller than $d$, i.e., $d_s < d$) following [78]. Let $e_{1:k}$ represent the sequence of all scores. Assume $j$ is the position of the first node in the maximum scoring pair, i.e., $j = argmax(e_{1:k})$. Now the update rule for the input to the next recursion will be:

$$h_i^{t+1} = \begin{cases} h_i^t & i < j \\ R(h_i^t, h_{i+1}^t) & i = j \\ h_{i+1}^t & i > j \end{cases} \tag{4}$$

The iterative application of this rule can achieve both tree-parsing (through $scorer$) and sequence composition (by application of $R$) simultaneously. Finally, we will have the root representation left which can be used as a sentence encoding (the intermediate non-terminal nodes can be also used if needed). Note that the framework we describe already incorporates the efficiency fixes introduced in [78]. Works [12, 77] prior to [78] made the scorer's input to be $R(h_i^t, h_{i+1}^t)$ which requires running expensive recursive cells for all possible node pair siblings (not just the chosen one) in parallel and they use no slicing. A problem with directly using the above framework is that the argmax for selecting $j$ is non-differentiable. So it becomes impossible to train the $scorer$. Various techniques have been used to address this problem - e.g., with autoencoder loss [88], gumbel softmax [62, 45] with Straight-through Estimation (STE) [4], or with SPIGOT [74]. Although not all of them have been exhaustively explored, STE gumbel softmax [12] which is one of the more popular approaches has been found to fail in structure-sensitive tasks [70, 38]. However, viewing the above family of approaches as a greedy search approach - opens up another way that is using simple beam search replacing argmax with a top-$k$ operator for beam search [77]. We discuss this below.

**Beam Search Tree RvNN:** Extending Greedy Search Tree RvNNs with beam search allows for keeping multiple hypotheses or beams. Each beam would be one possible induced tree structure with its node representations and a corresponding beam score (addition of the log-softmax of $e_j^t$ for each iteration $t$ in that beam). Setting up a beam-score-based softmax-based marginalization over the root representations of each beam in the end allows us to create a sentence vector representation: $\sum_i \frac{\exp(s_i)}{\sum_j \exp(s_j)} \cdot b_i$ where $b_i$ is the root representation of beam $i$ and $s_i$ is the beam score. The final softmax can allow competition in the top $B$ selected beams (where $B$ is the beam width) which can allow meaningful gradient signals to reach the scorer. Ray Chowdhury and Caragea [77] showed that this is enough to make the former framework succeed in ListOps and other logical inference tasks. We use the efficient variant Beam Tree RvNN (BT-RvNN), i.e., Efficient Beam Tree RvNN (EBT-RvNN) which is BT-RvNN built on the specific greedy framework described above rather than the prior frameworks [12] that we contrasted. Both BT-RvNN and EBT-RvNN use stochastic Top-$K$ [52]. More details on BT-RvNN can be found in [77] and more details on EBT-RvNN can be found in [78].

**Balanced Tree RvNNs:** Above, we focused on Binary (2-ary) Tree RvNNs. Here we consider a more general $k$-ary Tree RvNN. The non-terminal nodes of $k$-ary Tree RvNN can have at most $k$ children. Since we defined the recursive cell $R$ to have two parameters, we now introduce a generalized form of the recursive cell - $RK : \mathbb{R}^{k \times d_h} \to \mathbb{R}^{d_h}$. $RK$ takes a sequence of $k$ vectors as input and outputs a single vector. Essentially $RK$, by itself, operates like a sentence encoder. Next, we introduce a balanced tree RvNN model that uses a $k$-ary recursive cell.

We call the balanced tree variant of a $k$-ary Tree RvNN as Balanced $k$-ary Tree-RvNNs (BKT-RvNNs). Balanced Binary Tree RvNNs (BBT-RvNNs) is a special case of BKT-RvNN when $k = 2$. In each recursion, BKT-RvNNs make non-overlapping chunks of size $k$ (so we can also say $k$ is the "chunk size" henceforth) and then compress each chunk into a single vector using $RK$. This is done in every recursion until there is only one vector left - which becomes the root node representation, i.e., the final sentence encoding. Let us say, we have a sequence ($r_{1:n} \in \mathbb{R}^{n \times d_h}$) of the form below:

$$r_{1:n} = r_1, r_2, \ldots, r_n \tag{5}$$

Assume that $n$ is divisible by $k$ - if not, we can always use padding to extend the sequence length to make it divisible. We can then create a chunked sequence ($c_{1:\lfloor \frac{n}{k} \rfloor} \in \mathbb{R}^{\lfloor \frac{n}{k} \rfloor \times k \times d_h}$) as below:

$$c_1 = (r_1, r_2, \ldots, r_k), \ \ c_2 = (r_{k+1}, \ldots, r_{2k}), \ldots, \ \ c_{\lfloor \frac{n}{k} \rfloor} = \left( r_{(\lfloor \frac{n}{k} \rfloor - 1) \cdot k + 1}, r_{(\lfloor \frac{n}{k} \rfloor - 1) \cdot k + 2}, \ldots, r_n \right) \tag{6}$$

Each chunk ($c_i$) can be then processed independently (and simultaneously) by $RK : \mathbb{R}^{k \times d_h} \to \mathbb{R}^{d_h}$ to compress the $k$-sized chunks into single vectors:

$$p_1 = RK(c_1), p_2 = RK(c_2), \ldots, p_{\lfloor \frac{n}{k} \rfloor} = RK(c_{\lfloor \frac{n}{k} \rfloor}) \tag{7}$$

Thus the sequence of chunks, $c_{1:\lfloor \frac{n}{k} \rfloor} \in \mathbb{R}^{\lfloor \frac{n}{k} \rfloor \times k \times d_h}$ turns into a sequence of vectors $p_{1:\lfloor \frac{n}{k} \rfloor} \in \mathbb{R}^{\lfloor \frac{n}{k} \rfloor \times d_h}$. $p_{1:\lfloor \frac{n}{k} \rfloor}$ then becomes the input to the next recursion. We terminate the recursion when only one item is left in the sequence. Since in every recursion the sequence size is cut by $k$ times, the total recursions taken (until a sequence of size 1 is left) would be $\lceil \log_k n \rceil$ assuming $n$ as the original sequence length.

## 3 Recursion in Recursion

Here, we introduce our framework - Recursion in Recursion (RIR). RIR is essentially an implementation of BKT-RvNN. The question to ask when using BKT-RvNN is what to use to implement $RK$. In case of RIR, we use another Tree-RvNN to implement $RK$. This results in a two-level nested recursion as discussed below. We present a visualization in Figure 1 and explain the two levels below.

**Outer Recursion:** The outer recursion (outer RvNN) is the BKT-RvNN setup as discussed.

**Inner Recursion:** In RIR, the inner recursion is another RvNN (inner RvNN). In principle, we can use nearly anything for the inner RvNN or the $RK$ function in general - even a Transformer [94] or

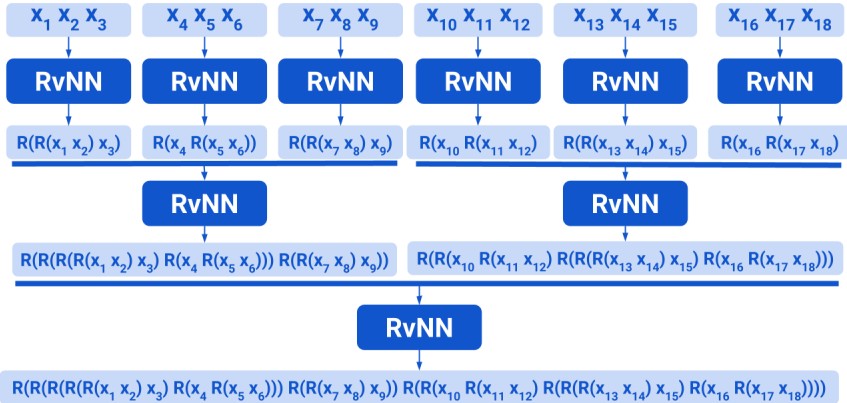

Figure 1: Visualization of the Recursion In Recursion (RIR) framework. The sequence $(x_1, x_2, \ldots, x_9)$ is the input sequence of vectors. The RvNN block indicates the inner RvNN. $R$ is the recursive cell within the inner RvNN. The inner RvNN block is working within a balanced $k$-ary Tree like structure (here $k = 3$) - effectively serving as the recursive cell within BKT-RvNN.

an RNN. Using an RNN makes the whole model similar to Sliced RNN [101]. Our main RIR-based model uses EBT-RvNN as the inner RvNN. We motivate this choice below.

**Motivation for EBT-RvNN:**

1. BT-RvNN performs quite well on logical inference and ListOps while maintaining reasonable natural language performance - a feature lacking in plain RNNs, Gumbel-Tree variants, or other simpler RvNNs [77]. EBT-RvNN is a more promising and efficient variant of BT-RvNN [78].

2. EBT-RvNN as an inner RvNN (after some adjustments to be discussed in the next section) offers us a clean interpretation of the RIR model as an approximation of EBT-RvNN searching in an additionally constrained space of tree structures (constrained by the outer balanced tree recursion). While other ListOps competent models can be also forcibly used as an inner RvNN [13, 82], there will be less of an interpretation of their modeling choices under RIR.

**Time Complexity Analysis:** The time complexity of the outer RvNN (i.e. BKT-RvNN) in RIR (if the inner RvNN were $\Theta(1)$) is $\Theta(log_k(n))$ following what we already discussed. Let us say that the time complexity of the inner RvNN is some $\Theta(f(n))$ if $n$ is the input sequence length. Given these factors, the overall time complexity of an RIR model will be $\Theta(\log_k(n)f(min(k,n)))$. While inner RvNNs like EBT-RvNN can be slow by itself, within the RIR framework its recursive depth is now bound by the chunk size $k$ instead of the sequence length $n$. The chunk size $k$ is a hyperparameter that is preset as a constant before training. It remains constant during training no matter how big the sequence length $n$ becomes. As such although the inner RvNN can be expensive it now becomes bound by constant factors during training. This makes the RIR model with EBT-RvNN as inner RvNN much more efficient than EBT-RvNN itself. The same is also true for other possible implementations of an inner RvNN within the RIR framework.

However, the inner recursion with EBT-RvNN can be still relatively slow and add large overheads. Thus, the bigger the $k$, the slower the overall model will be. On the other hand, lower $k$ pushes the model closer to BBT-RvNN and makes it faster. When $k = 2$, the model would just collapse into BBT-RvNN, and when $k = \infty$, the model reduces into just EBT-RvNN.

### 3.1 Beam Alignment

**Context:** There are a few challenges to naively integrating EBT-RvNN in the RIR framework. First, note that we initially defined $RK$ as of the form $RK : I\!R^{k \times d_h} \to I\!R^{d_h}$. If we keep it that way then we have to marginalize the beams at the end of every inner recursive loop to get a single vector output instead of beams of vectors. This can make it harder to extract discrete structures from the

beam search process if needed for interpretability or for sending top-down attention signals for token contextualization (as used in [78] with parent attention). Also, it is in conflict with the second motivation for using EBT-RvNN. Moreover, keeping the beams active instead of marginalizing them every so often can allow for more robustness and recovery from error [20]. Now, if the beams are kept active, there are two main changes.

1. The input sequence in every outer recursion would belong to $\mathbb{R}^{b \times n \times d_h}$ where $b$ is the beam size. The output sequence after the outer recursion should be in $\mathbb{R}^{b \times \lfloor \frac{n}{k} \rfloor \times d_h}$. In addition, we need to also maintain an input of beam scores $\in \mathbb{R}^b$ and an output of the same form.

2. Each chunk that is fed to $RK$ should be now of the form $\mathbb{R}^{b \times k \times d_h}$. In addition, the beam scores $\in \mathbb{R}^b$ are also fed to $RK$. The output of $RK$ should be in the form of a tuple: $(BS, BR)$ where $BS \in \mathbb{R}^b$ represents the list of beam scores for each beam and $BR \in \mathbb{R}^{b \times 1 \times d_h}$ is the list of corresponding chunk-level sentence encodings[2] for each beam. Overall $RK$ is to be redefined as: $RK : \mathbb{R}^b \times \mathbb{R}^{b \times k \times d_h} \rightarrow \mathbb{R}^b \times \mathbb{R}^{b \times 1 \times d_h}$. The first input argument represents the input beam scores, the second represents the beam of chunks.

**Problem:** Now, the output of each $RK$ is a sequence $\in \mathbb{R}^{b \times 1 \times d_h}$ and beam scores $\in \mathbb{R}^b$. Since there would be $\lfloor \frac{n}{k} \rfloor$ chunks (if the input sequence length to that level of outer recursion is $n$), there will be simultaneous $\lfloor \frac{n}{k} \rfloor$ sequences $\in \mathbb{R}^{b \times 1 \times d_h}$ and similarly, $\lfloor \frac{n}{k} \rfloor$ lists of beam scores $\in \mathbb{R}^b$. The challenge is to answer how we get a single sequence $\in \mathbb{R}^{b \times \lfloor \frac{n}{k} \rfloor \times d_h}$ and a single list of beam scores $\in \mathbb{R}^b$ from this circumstance to serve as inputs to the next outer recursion.

For simplicity, let us say $\lfloor \frac{n}{k} \rfloor = 2$ and as such, we have two chunk outputs: $(BS_1, BR_1)$ and $(BS_2, BR_2)$. The question that is now raised is: How do we combine the different beams of the different output tuples for the input to the next outer recursion?

**String-it Solution:** One answer can be to just string together (concatenate) the corresponding beams in $BR_1$ and $BR_2$ as: $BR_{new} = \text{concatenate}([BR_1, BR_2], dim = 1)$. In other words, we concatenate $BR_1[i]$ with $BR_2[i]$ for all $i \in \{0, 1, \ldots, b - 1\}$. Here, $BR_{new} \in \mathbb{R}^{b \times 2 \times d}$. The beam scores can be summed up for accumulation: $BS_{new} = BS_1 + BS_2$.

**Problem with the String-it Solution:** The choice above is arbitrary. While concatenating the top scoring beams (the first beam of every $BR$) can be motivated, there is no reason to disallow beams from different ranks (say $BR_1[i]$ and $BR_2[j]$ where $i \neq j$) from being concatenated together especially if they form a unique beam with higher combined scores than others.

**Beam Alignment Solution:** Given the above problem, we introduce our solution - the Beam Alignment technique. In this technique, we apply a simple stochastic approach following the idea that "higher scoring beams should have higher chance to exist in the combined beams"[3]. Based on this idea, we simply transform each $(BS_l, BR_l)$ into some $(BS'_l, BR'_l)$ (we explain this below) and then apply the string-it solution on the transformed outputs $(BS'_l, BR'_l)$ to get the input for the next outer recursion. The transformation works as follows:

$$\begin{aligned} idx_l = \text{sample}(&\text{population} = [0, 1, \ldots, b - 1], \\ &\text{distribution} = \text{normalize}(BS_l), \\ &\text{times} = b) \end{aligned} \quad (8)$$

That is, first, we sample the indices ($idx_l$) of beams based on their corresponding scores ($BS_l$). The scores $BS_l$ can be normalized with softmax. We sample $b$ times with repetition. The next steps are:

$$BS'_l = [BS_l[j] \text{ for } j \text{ in } idx_l]; \quad BR'_l = [BR_l[j] \text{ for } j \text{ in } idx_l] \quad (9)$$

Thus, beams with higher scores will be sampled more times and in turn will have a higher chance to be present in $BS'_l$ and $BR'_l$, and in turn, more likely to be present in the stringed-together beams $BS_{new}$ and $BR_{new}$ after applying the string-it solution. As such, the high-level idea of increasing

---

[2]That is the beams of sentence encoding for the chunk input $\in \mathbb{R}^{B \times k \times d_h}$ as created by the inner EBT-RvNN processor.

[3]Alternatively one could also try to search for an optimal combination of beams but that can increase compute and may end up losing diversity in the stringed beams.

the chance of higher scoring beams is implemented[4]. One additional detail: in practice, we enforce $BS_{new}[0]$ and $BR_{new}[0]$ to be the concatenation of the top beams from the pre-transformed materials $(BS_l, BR_l)$ so that the best possible stringed beam is guaranteed to be present. We provide additional intuition on beam alignment with added visualization in Appendix I.

## 3.2 Pre-Chunk Processing

The chunking used in BKT-RvNN does not take into account the input content. This can create bad chunks where the information necessary to process one chunk is in another chunk. For example, imagine what happens if we split `MAX(1,2,3,4)` into three chunks `[MAX(1`, `[2,3,4]`, and `[)]` (where the square brackets indicate chunk boundaries). There would be no information initially for the later chunks to work on. To mitigate that we use an initial (single) linear RNN layer bidirectionally to propagate information outside chunks before starting the RIR loop. Specifically, we use an S4D with Gated Linear Units [30] (described more in Appendix J). Moreover, earlier literature has often used a recurrent layer to initialize the non-terminals before running their Tree-RvNN [12, 84] for more contextualized parsing decisions. So the S4D layer can be seen as serving that purpose too.

## 3.3 Practical Tips and Points

The chunk size ($k$) for BKT-RvNN can effectively work as a trade-off parameter in the RIR framework. Lower chunk size can make the whole pipeline faster by having less work to do for the heavy inner-RvNN, but for the same reason the point of using EBT-RvNN (instead of a simpler RNN model) would start to get lost as we lower the chunk size. In Appendix C, we show that lowering chunk size drastically deteriorates the performance of our model in ListOps. A practical option is to set the chunk size as high as possible while meeting one's practical computational need. We generally set $k = 30$ for our purposes. We also discuss theoretical ways to extend the framework for language modeling in Appendix H.

**RIR Inference:** As mentioned before, EBT-RvNN runs within the RIR framework and can be interpreted as just EBT-RvNN with some additional constraints in its tree-search space. Thus, during inference, we can just run EBT-RvNN as it is without RIR given that the core model remains the same - now just being allowed to cover more space and gain enhanced performance. By default, we generally indeed do that during inference even if the EBT-RvNN model is trained under RIR however using RIR inference generally works just as well for less structure-sensitive tasks. We discuss more about this decision in Appendix D.

# 4 Experiments and Results

## 4.1 Model Nomenclature

**RNN-GRC** represents an RNN implemented with Gated Recursive Cell (GRC) as the recursive cell $R$ (GRC was introduced in [82] and discussed in Appendix J); **CRvNN** refers to Continuous Recursive Neural Network [13]; **OM** refers to Ordered Memory [82]; **BT-GRC** refers to **BT-RvNN** implemented with GRC [77]; **BT-GRC OS** refers to BT-GRC combined with OneSoft (OS) Top-$K$ function [77]; **EBT-GRC** refers to EBT-RvNN model with GRC; **S4D** refers to stacked bidirectional S4D model implemented similar to S4D-inv in [30]; **MEGA** refers to a hybrid model combining Transformers and an SSM based on exponential moving average as introduced in [60]; **BBT-GRC** refers to BBT RvNN [84] but with GRC (it is intended to be a more basic baseline and does not have S4D as an initial layer), **RIR-GRC** refers to an RIR-based model using a recurrent GRC cell for the inner RvNN (it can be thought to be similar to sliced RNN [101]), **RIR-EBT-GRC** is our main proposed model. Both of the latter models use an S4D layer before starting the main RIR loop.

---

[4]The overall strategy can create some repetitions of beams. However, we do not think this should happen too often because the score distributions tend to have high entropy in practice. Moreover, the original BT-RvNN framework already allows repetitions and still works well. Trying to completely remove repetitions can require some tricky operation that takes up resources without much return. We leave it for future research for further analysis on potential ways to avoid repetitions.

Table 1: Empirical time and (peak) memory consumption for various models on an RTX A6000. Ran on 100 ListOps data with batch size 1 and the same hyperparameters as used on ListOps on various sequence lengths. For the splits of lengths $200 - 1000$ we use the data shared by Havrylov et al. [38]; for the $1500 - 2000$ split we sample from the training set of LRA listops [92]

| Model | Sequence Lengths | | | | | | | |
| | $200 - 250$ | | $500 - 600$ | | $900 - 1000$ | | $1500 - 2000$ | |
| | Time (min) | Memory (GB) | Time (min) | Memory (GB) | Time (min) | Memory (GB) | Time (min) | Memory (GB) |
|---|---|---|---|---|---|---|---|---|
| RNN-GRC | 0.22 | **0.02** | 0.54 | **0.02** | 1.3 | 0.03 | 2.5 | **0.04** |
| MEGA | 0.06 | 0.07 | 0.05 | 0.16 | 0.05 | 0.29 | 0.05 | 0.72 |
| S4D | 0.04 | 0.03 | 0.05 | 0.05 | 0.04 | 0.07 | 0.05 | 0.14 |
| BBT-GRC | **0.02** | **0.02** | **0.02** | **0.02** | **0.03** | **0.02** | **0.03** | **0.04** |
| RIR-GRC | 0.06 | **0.02** | 0.08 | 0.03 | 0.09 | 0.04 | 0.1 | 0.07 |
| ListOps Competitive | | | | | | | | |
| OM | 8.0 | 0.09 | 20.6 | 0.21 | 38.2 | 0.35 | 76.6 | 0.68 |
| CRvNN | 1.5 | 1.57 | 4.3 | 12.2 | 8.0 | 42.79 | OOM | OOM |
| BT-GRC | 1.1 | 1.71 | 2.6 | 9.82 | 5.1 | 27.27 | OOM | OOM |
| BT-GRC OS | 1.4 | 2.74 | 4.0 | 15.5 | 7.1 | 42.95 | OOM | OOM |
| EBT-GRC | 1.2 | 0.19 | 3.2 | 1.01 | 5.5 | 2.78 | 10.5 | 10.97 |
| RIR-EBT-GRC | **0.1** | **0.07** | **0.2** | **0.14** | **0.3** | **0.23** | **0.3** | **0.43** |

Table 2: Accuracy on ListOps-O. For our models we report the median of 3 runs. Our models were trained on lengths $\leq 100$, depth $\leq 20$, and arguments $\leq 5$. We bold the best results that do not use gold trees. Subscript represents standard deviation. We use the original training set [70] with the length generalization splits from Havrylov et al. [38], the argument generalization splits from Ray Chowdhury and Caragea [77], and the LRA test set from Tay et al. [92]. As an example, $90_1 = 90 \pm 0.1$

| Model | near-IID | Length Gen. | | | Argument Gen. | | LRA |
| (Lengths) | $\leq 1000$ | 200-300 | 500-600 | 900-1000 | 100-1000 | 100-1000 | 2000 |
| (Arguments) | $\leq 5$ | $\leq 5$ | $\leq 5$ | $\leq 5$ | 10 | 15 | 10 |
|---|---|---|---|---|---|---|---|
| MEGA | $75.45_{2.6}$ | $45.2_{1.5}$ | $31.7_{14}$ | $24.7_{36}$ | $39.5_{33}$ | $33.8_{59}$ | $25.8_{49}$ |
| S4D | 72.33 | 31 | 20.85 | 14.7 | 27.3 | 22.76 | 17.4 |
| BB-Tree GRC | 59.18 | 43.6 | 40.4 | 31.5 | 45.35 | 44.5 | 38.25 |
| RIR-GRC | 78.33 | 41.75 | 35.55 | 32.3 | 43.8 | 42.75 | 40.05 |
| OM | **99.9** | 99.6 | 92.7 | 76.9 | **84.15** | 75.05 | **80.1** |
| CRvNN | 99.82 | 99.5 | 98.5 | 98 | 65.45 | 45.1 | 55.38 |
| BT-GRC OS | **99.9** | 99.5 | 99 | 97.2 | 76.05 | 67.9 | 71.8 |
| EBT-GRC | **99.9** | **99.9** | **99.4** | **99.5** | 82.95 | **79** | 79.5 |
| RIR-EBT-GRC | 99.72 | 99.15 | 98.25 | 97.1 | 74.9 | 49.55 | 61.65 |
| $-$S4D | 99.78 | 99.15 | 98.87 | 98.6 | 63.95 | 47.73 | 48.25 |
| $-$Beam Align | 98.83 | 91.75 | 79.05 | 68.8 | 59.55 | 43.65 | 40.75 |

## 4.2 Results

For more details on the architectures and experiments, see Appendix A. There are also more ablations in Appendix C.

**Efficiency Analysis:** In Table 1, we compare the empirical time-memory trade-offs of the most relevant Tree-RvNN models. We use CRvNN in the no halting mode as [77][5]. As can be seen, our RIR-EBT-GRC is multiple times faster than the original BT-GRC models and even EBT-GRC

---

[5]Otherwise it fails to learn to halt in the small training set and starts to halt too quickly.

Table 3: Mean accuracy and standard deviation on the Logical Inference for $\geq 8$ number of operations after training on samples with $\leq 6$ operations. Our models were run 3 times on different seeds. Subscript represents standard deviation. As an example, $90_1 = 90 \pm 0.1$

| Model | Number of Operations | | | | |
|---|---|---|---|---|---|
| | 8 | 9 | 10 | 11 | 12 |
| MEGA | $89.7_{16}$ | $83.62_{8.8}$ | $76.9_{18}$ | $70.45_{35}$ | $63.85_{36}$ |
| S4D | $87.6_{20}$ | $79.4_{36}$ | $72.32_{32}$ | $65.01_{53}$ | $59.48_{38}$ |
| BBT-GRC | $77.7_{15}$ | $72.5_{31}$ | $67.4_{11}$ | $64.0_{14}$ | $57.5_{12}$ |
| RIR-GRC | $86_{5.6}$ | $64.89_{8.9}$ | $47.35_{27}$ | $46.45_{26}$ | $45.92_{25}$ |
| OM | $\mathbf{97.5_{1.6}}$ | $\mathbf{96.74_{1.4}}$ | $94.95_2$ | $93.9_{2.2}$ | $93.36_{6.2}$ |
| BT-GRC OS | $97.03_{1.4}$ | $96.49_{1.9}$ | $95.43_{4.5}$ | $\mathbf{94.21_{6.6}}$ | $\mathbf{93.39_{1.5}}$ |
| EBT-GRC | $97.12_3$ | $96.5_{3.1}$ | $94.95_{1.5}$ | $93.87_{7.4}$ | $93.04_{6.7}$ |
| RIR-EBT-GRC | $96.84_{1.4}$ | $96.05_{1.3}$ | $\mathbf{95.45_{3.4}}$ | $\mathbf{94.21_{3.7}}$ | $93.36_{11}$ |

Table 4: Accuracy on LRA [92]. * represents that the results are copied from [30] and † represents that the results are copied from [60]. For our models we report the mean of 2 runs. ListOpsMix combines the training set of ListOps-O and ListOps LRA; it keeps the dev set and test set of LRA. Subscript represents standard deviation. As an example, $90_1 = 90 \pm 0.1$

| Model | LRA | | | ListOpsMix |
|---|---|---|---|---|
| | ListOps | Text | Retrieval | |
| MEGA† | $\mathbf{63.14}$ | $\mathbf{90.43}$ | $\mathbf{91.25}$ | — |
| S4D* | $60.18_{3.5}$ | $87.34_2$ | $91.09_{.1}$ | — |
| S4D | $59.3_0$ | $88.18_{0.4}$ | $91.17_{.6}$ | $59.05_{2.5}$ |
| BBT-GRC | $55.15_1$ | $85.69_{0.7}$ | $90.78_{1.6}$ | $55.7_{15}$ |
| RIR-GRC | $20.18_{5.7}$ | $86.894_{1.3}$ | $88.24_{10}$ | $58.95_{6.5}$ |
| RIR-EBT-GRC | $59.0_4$ | $87.97_{1.6}$ | $88.99_{5.2}$ | $64.1_{3.5}$ |
| -S4D | $59.53_1$ | $88.13_{1.3}$ | $89.74_{3.9}$ | $\mathbf{70.43_{90}}$ |

while using much less memory. As expected, because of still using a heavy inner loop model, it is not as competitive resource-wise against S4D, MEGA, BBT-GRC, or RIR-GRC. However, in our experiments, we find these competitors to fail on ListOps and logical inference whereas RIR-EBT-GRC still works. We present a pareto frontier analysis in the Appendix F.

**List Operations (ListOps):** Here we discuss the results of Table 2. This table is based on the original ListOps dataset [70]. To disambiguate it from the ListOps used in LRA [92], we call this one ListOps-O. Similar to [13], we filtered all sequences of length $> 100$ from ListOps-O training set. ListOps-O involves training on much shorter samples than those in LRA ListOps but ListOps-O can be challenging in its own right because this allows investigating length-generalization prowess of neural networks. ListOps requires solving hierarchically nested lists of mathematical operations that neural networks, barring a few exceptions [38, 13, 82, 77], generally struggle to solve. The different splits test the models in different out-of-distribution settings, e.g., unseen lengths, unseen number of arguments, or both. As can be seen from Table 2, RIR-EBT-GRC does fairly well in length-generalization (still getting $\geq 90\%$) despite being perturbed within the RIR framework for efficiency. Its argument generalization performance is, however, hampered. Removing the S4D layer (used for pre-chunk preprocessing) from RIR-EBT-GRC (the $-$S4D row) does not actually harm length generalization but seems to harm argument generalization. Removing beam alignment from RIR-EBT-GRC (and just keeping the string-it solution) substantially deteriorates the performance showing the effectiveness of beam alignment (see $-$Beam Align row). MEGA, S4D and RIR-GRC perform reasonably in the near-IID[6] setting compared to LSTMs and Universal Transformers [82] but does not come close to more powerful RvNNs. BBT-GRC also does not work well here.

---

[6]It is "near" IID because it has some longer sequence data than what is present in the training set. But generally, most data are on the shorter end.

**Logical Inference:** In Table 3, we show the results of our models in a formal logical inference task [6]. This is another task where only stronger Tree-RvNN based models have been able to perform well. In this task the models are trained on samples with $\leq 6$ operators and we test their generalization performance on higher operators (similar to [93]). We find that RIR-EBT-GRC can keep up well with the state-of-the-art whereas others like MEGA, S4D, BBT-GRC, or RIR-GRC cannot.

**LRA and ListOpsMix:** In Table 4 we explore the NLP subset of LRA tasks. We find that BBT-GRC can already perform quite well in LRA without any special initialization (note that BBT-GRC does not use S4D), outperforming all non-hybrid Transformer models [18, 106]. RIR-EBT-GRC also shows high competence in LRA. It starts to get a bit slow on the Retrieval task so we reduced the chunk size to 5 which might reduce its performance there. While RIR-EBT-GRC uses S4D for pre-chunk processing, we also run a variant without any S4D (see row $-$S4D) and it performs just as well or better. This shows that the S4D layer is not playing a big role here. Curiously enough, RIR-EBT-GRC does worse on ListOps LRA test in the in-distribution setting (in Table 4), than it does in the OOD setting trying to generalize from shorter sequences with lower arguments in Table 2. We hypothesize that this might be because RIR-EBT-GRC learns more easily lower-length samples. To investigate whether RIR-EBT-GRC can gain more on LRA when samples of shorter length are present, we created another ListOps dataset - ListOpsMix which combines the ListOps-O training set and the ListOps LRA training set (ListOps LRA dev and test sets are used for validation and testing, respectively). As we can see RIR-EBT-GRC (with or without S4D) is particularly better at utilizing extra lower-length data than other models; however, it still underperforms compared with what EBT-GRC/OM can get with training purely on ListOps-O.

**Other Tasks:** We also ran our models in a number of other natural language tasks, e.g., natural language inference and sentiment classification. The results are in Appendix E. Generally RIR-EBT-GRC performs similarly to other RvNN-based models.

## 5 Conclusion

In this paper, we introduce the Recursion In Recursion (RIR) framework and beam alignment to make Efficient Beam-Tree RvNNs (EBT-RvNN) more scalable. We also show the power of more basic models like BBT-RvNNs on LRA. Even though they do not outperform S4D or some of the newer state-of-the-art [60, 86, 85, 24], BBT-RvNNs are coming from a completely orthogonal family of models which, in recent times, have received limited attention. Moreover, RIR-EBT-RvNNs provide a better trade-off in maintaining competitive performance in LRA text-related tasks and robust length generalization in structure-sensitive tasks as shown in the Pareto Frontier plots in Appendix F. We intend our results to inspire further exploration of these models. We provide an extended discussion with related works in Appendix G.

## 6 Limitations

Our main method, RIR-EBT-GRC, although much more computationally efficient than other ListOps-competent RvNNs [82, 13, 77], is still quite a bit slower compared to BBT-GRC, RIR-GRC, MEGA, and S4D. Moreover, although it serves as a non-linear recursive model that can both scale to training on $\geq 1000$ sequence lengths and also length-generalize on ListOps and Logical Inference (from shorter samples), it has still difficulties learning from longer sequences.

## 7 Acknowledgments

This research is supported in part by NSF CAREER award #1802358, NSF IIS award #2107518, and UIC Discovery Partners Institute (DPI) award. Any opinions, findings, and conclusions expressed here are those of the authors and do not necessarily reflect the views of NSF or DPI. We thank our anonymous reviewers for their constructive feedback.

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

## A  Organization

In Section B, we describe the settings of all the tasks and datasets that we used for model evaluation. In Section C, we provide an extended ablation of RIR-EBT-GRC. In Section D, we discuss in more detail the RIR inference strategy. In Section E, we provide additional results on LRA, sentiment classification, natural language inference, and paraphrase detection. In Section F, we provide a Pareto Frontier analysis of different trade-offs for different models. In Section G, we present an extended survey of related work. In Section H, we discuss ways to use RIR-based models for language modeling. In Section I, we provide more intuition and visualization for beam alignment. In Section J, we detail our architecture setup including the S4D. In Section K, we describe our hyperparameters.

## B  Details on Evaluation Tasks

**ListOps:** ListOps was originally introduced by Nangia and Bowman [70]. It is a task for solving nested lists of mathematical operations. It is a 10-way classification task. Like Chowdhury and Caragea [13], we train our models on the original training set with all samples $\geq 100$ sequence lengths filtered away. We use the original development set for validation. We test on the original test set (near-IID split); the length generalization splits from Havrylov et al. [38] that include samples of much higher lengths; the argument generalization splits from Ray Chowdhury and Caragea [77] that involve an unseen number of maximum arguments for each operator; and the LRA split (which has both higher sequence length and higher argument number) from Tay et al. [92].

**Logical Inference:** Logical Inference was introduced by Bowman et al. [6]. This task involves classifying fine-grained inferential relations between two given sequences in a form similar to that of formal sentences of propositional logic. Similar to Tran et al. [93], our models were trained on splits with logical connectives $\leq 6$. We show the results in OOD test sets with logical connections 8-12. We use the same splits as Shen et al. [82], Tran et al. [93], Chowdhury and Caragea [13].

**Long Range Arena (LRA):** LRA is a set of tasks designed to evaluate the capacities of neural models for modeling long-range dependencies [92]. We evaluate on the language-related subset of tasks - ListOps LRA, Text LRA (character-level IMDB [61]), and Retrieval LRA (character-level document pair matching task).

**SST5:** SST5 is a fine-grained 5-way sentiment classification task introduced in Socher et al. [89]. We use the original splits.

**IMDB:** IMDB is a binary sentiment classification task from Maas et al. [61]. We use the same train, validation, and IID test sets as created in Ray Chowdhury and Caragea [77]. We also use the contrast set Gardner et al. [25] and counterfactual set Kaushik et al. [49] as additional test splits.

**QQP:** QQP[7] is a task of classifying whether two given sequences in a pair are paraphrases or not. As standard Wang et al. [95], we randomly sample $10,000$ samples for validation and IID test set such that for each split 5000 samples are maintained to be paraphrases and the other 5000 are maintained to be not paraphrases. We also use the adversarial test sets $\text{PAWS}_{QQP}$ and $\text{PAWS}_{WIKI}$ form Zhang et al. [102].

**SNLI:** SNLI [5] is a natural language inference (NLI) task. It is a 3-way classification task to classify the inferential relation between two given sequences. We use the same train, development, and IID test set splits as in Chowdhury and Caragea [13]. Any data with a sequence of length $\geq 150$ is filtered from the training set for efficiency. We also use additional test set splits for stress tests. We use the hard test set split from Gururangan et al. [34], the break test set from Glockner et al. [27], and the counterfactual test set from Kaushik et al. [49].

**MNLI:** MNLI [96] is another NLI dataset which is similar to SNLI in format. We use the original development sets (match and mismatch) as test sets. We filter away all data with any sequence length $\geq 150$ from the training set. Our actual development set is a random sample of $10,000$ data points from the filtered training set. As additional testing sets, we use the development set of Conjunctive NLI (ConjNLI) [80] and a few of the stress sets from Naik et al. [69]. These stress test sets include - Negation Match (NegM), Negation Mismatch (NegMM), Length Match (LenM), and Length Mismatch (LenMM). NegM and NegMM add tautologies containing "not" terms - this

---

[7] https://data.quora.com/First-Quora-Dataset-Release-QuestionPairs

Table 5: Extended Ablation on ListOps-O. For our models, we report the median of 3 runs. Our models are trained on lengths $\leq 100$, depth $\leq 20$, and arguments $\leq 5$. We bold the best results that do not use gold trees. We use the original training set [70] with the length generalization splits from Havrylov et al. [38], the argument generalization splits from Ray Chowdhury and Caragea [77], and the LRA test set from Tay et al. [92]. Subscript represents standard deviation. As an example, $90_1 = 90 \pm 0.1$

| Model | near-IID | Length Gen. | | | Argument Gen. | | LRA |
|---|---|---|---|---|---|---|---|
| (Lengths) | $\leq 1000$ | 200-300 | 500-600 | 900-1000 | 100-1000 | 100-1000 | 2000 |
| (Arguments) | $\leq 5$ | $\leq 5$ | $\leq 5$ | $\leq 5$ | 10 | 15 | 10 |
| RIR-GRC | 78.33 | 41.75 | 35.55 | 32.3 | 43.8 | 42.75 | 40.05 |
| RIR-CRvNN | 89.15 | 40.45 | 35.2 | 28.7 | 45.55 | 43.4 | 36.65 |
| RIR-OM | 97.62 | 75.7 | 42.20 | 29.7 | 62.55 | 47.45 | 36.20 |
| RIR-EBT-GRC | 99.72 | **99.15** | 98.25 | 97.1 | **74.9** | 49.55 | **61.65** |
| $-$S4D | **99.78** | **99.15** | **98.87** | **98.6** | 63.95 | 47.73 | 48.25 |
| $-$Beam Align | 98.83 | 91.75 | 79.05 | 68.8 | 59.55 | 43.65 | 40.75 |
| $+$Random Align | 99.14 | 96.89 | 93.3 | 86.4 | 67.2 | 53.2 | 55.05 |
| $+$RIR inference | 87.53 | 46.5 | 42.2 | 36.9 | 47.4 | 44.1 | 39.75 |
| (beam 5) | 99.4 | 97.8 | 94.3 | 89.0 | 65.6 | 53.15 | 48.25 |
| (chunk 20) | 98.9 | 91.75 | 77.85 | 63.7 | 66.8 | **55.7** | 50.5 |
| (chunk 10) | 89.64 | 47.5 | 47.35 | 39.9 | 45.9 | 37.75 | 33.25 |

can bias the models to classify contradiction as the inferential relation because the training set has spurious correlations between the existence of "not" related terms and the class of contradiction. LenM and LenMM add tautologies to artificially increase the lengths of the samples without changing the inferential relation class.

## C Extended Ablation

In Table 5, we show an extended ablation of RIR-EBT-GRC. As we can see, alternatives to EBT-GRC like CRvNN or Ordered Memory (OM) do not work as well within the RIR framework (see RIR-CRvNN or RIR-OM row) although they can be still better than RIR-GRC. We also ran another ablation for Beam Align where we replaced it with another method - Random Align. In Random Align, instead of sampling beams based on beam scores, we sample them based on a uniform distribution. See the "$-$Random align" row for the result of this method. As can be seen from the table, it does not perform as well. This shows the effectiveness of the high-level idea behind Beam Alignment over another control test. As discussed in the main paper, we generally switch out of the RIR framework during inference. "$+$RIR inference" shows the result of using the RIR framework during inference too. The results for "$+$RIR inference" worsen significantly. Given the efficiency of RIR-EBT-GRC, we also increase its default beam size to 7 from the original beam size of 5 as used in EBT-GRC and BT-GRC models. The row "(beam 5)" shows the result of using beam size 5 with RIR-EBT-GRC. The results worsen a bit. RIR-EBT-GRC uses a chunk size of 30. We also show the performance degradation with lower chunk sizes - see "(chunk 20)" and "(chunk 10)". These results worsen as is consistent with our discussion in the main paper.

## D RIR Inference

The RIR strategy enforces an outer balanced-tree structural restriction on the search space that is generally unsuited for structure-sensitive tasks (and merely done for efficiency). However, the existence of data samples with sequence length less than or equal to chunk size leads to some non-RIR training instances which helps teach the inner recursion to learn to length-generalize. Thus, testing without RIR inference (i.e., running the inner recursion model in the full input) can still lead to length generalization and sometimes be even better due to the lack of balanced tree enforcement.

Table 6: Mean accuracy and standard deviation on SST5 [89] and IMDB [61]. **Count.** represents counterfactual test split from Kaushik et al. [49] and **Cont.** represent contrast test split from Gardner et al. [25] We bold the best results. Our models were run 3 times on different seeds. Subscript represents standard deviation. As an example, $90_1 = 90 \pm 0.1$

| Model | SST5 IID | IMDB IID | Cont. | Count. |
|---|---|---|---|---|
| CRvNN | $51.75_{11}$ | $91.47_{1.2}$ | $\mathbf{77.80_{15}}$ | $\mathbf{85.38_{3.5}}$ |
| OM | $52.30_{2.7}$ | $\mathbf{91.69_{0.5}}$ | $76.98_{5.8}$ | $83.68_{7.8}$ |
| BT-GRC | $\mathbf{52.32_{4.7}}$ | $91.29_{1.2}$ | $75.07_{29}$ | $82.86_{23}$ |
| BT-GRC OS | $51.92_{7.2}$ | $90.86_{9.3}$ | $75.68_{21}$ | $84.77_{11}$ |
| EBT-GRC | $52.22_1$ | $91.47_{1.2}$ | $76.16_{17}$ | $84.29_{12}$ |
| BBT-GRC | $50.98_{3.4}$ | $90.57_{1.1}$ | $76.37_{7.5}$ | $\mathbf{85.38_{1.35}}$ |
| RIR-GRC | $50.65_{2.3}$ | $90.60_{2.5}$ | $77.53_{30}$ | $85.04_{14}$ |
| RIR-EBT-GRC | $51.01_{7.9}$ | $90.86_{4.6}$ | $74.25_{14}$ | $82.92_{9.5}$ |

Thus, during inference, we have a choice of removing the RIR structure from RIR-EBT-GRC and employing it just like EBT-GRC. We refer to this inference mode as "non-RIR inference". If we make no changes to RIR-EBT-GRC during inference, we refer to it as "RIR inference".

**Cost of non-RIR inference:** Non-RIR inference can be multiple times slower than RIR inference. Moreover, in most tasks (including LRA), RIR inference works as well as non-RIR inference. So in most cases, RIR inference is a "win-win" choice (several times faster and with similar accuracy).

**Gain of non-RIR inference:** When a model is working in a structure-sensitive context (such as ListOps-O task), it may find it difficult to learn to length generalize as well through an RIR format. Here, non-RIR inference works much better as seen in the ablation (Table 2)). If RIR training fails to make the model learn the task well, then even for structure-sensitive tasks the difference between the accuracy of RIR inference and non-RIR inference can collapse (it happens for ListOps LRA). RIR inference in structure-sensitive tasks still performs better than MEGA or S4D.

Generally, we can take the choice of using RIR inference as a hyperparameter - the validation metric in the first 5 epochs clues us in which inference strategy to use. For simplicity, we do not tune this choice and by default only use non-RIR inference except for some LRA tasks (Text LRA and Retrieval LRA) where non-RIR inference becomes too slow. For instance, in Text LRA, RIR inference takes around $\sim 7$ minutes whereas non-RIR inference takes around $\sim 6$ hours. We find non-RIR inference to mainly make a substantial difference in accuracy in case of ListOps-O and Logical Inference.

# E    Additional Results

We show additional results of our models in sentiment classification (SST5, IMDB) in Table 6 and in natural language inference (MNLI, SNLI) and paraphrase detection in Table 7. Like prior models [82, 13, 77], the natural language results here are mixed. RIR-EBT-GRC does not work well on sentiment classification (it may require more careful tuning), but otherwise keeps up well in sentence-pair matching task - generally even outperforming EBT-GRC. BBT-GRC sometimes does surprisingly well in some OOD sets or stress sets - like the counterfactual set in IMDB and the Break test set from SNLI. We also present an extended comparison in LRA with other competitive models in Table 8. While in the paper we mainly focus on text-based tasks, in theory, RIR-EBT-GRC can be also used for vision-processing tasks. We generally found that we can get greater than $60\%$ in Sequential CIFAR (LRA Image task). Although that is much lower than SSM-related or Long Convolution models, it still outperforms any pure Transformer-based models. However, in our initial experiments, RIR-EBT-GRC fails in pathfinder tasks. There is more room to explore this area and perhaps look for better ways of creating synergy between SSM-based models and RIR-EBT-GRC. We did not find any notable difference in convergence speed (number of epochs taken) between RIR-EBT-GRC and EBT-GRC.

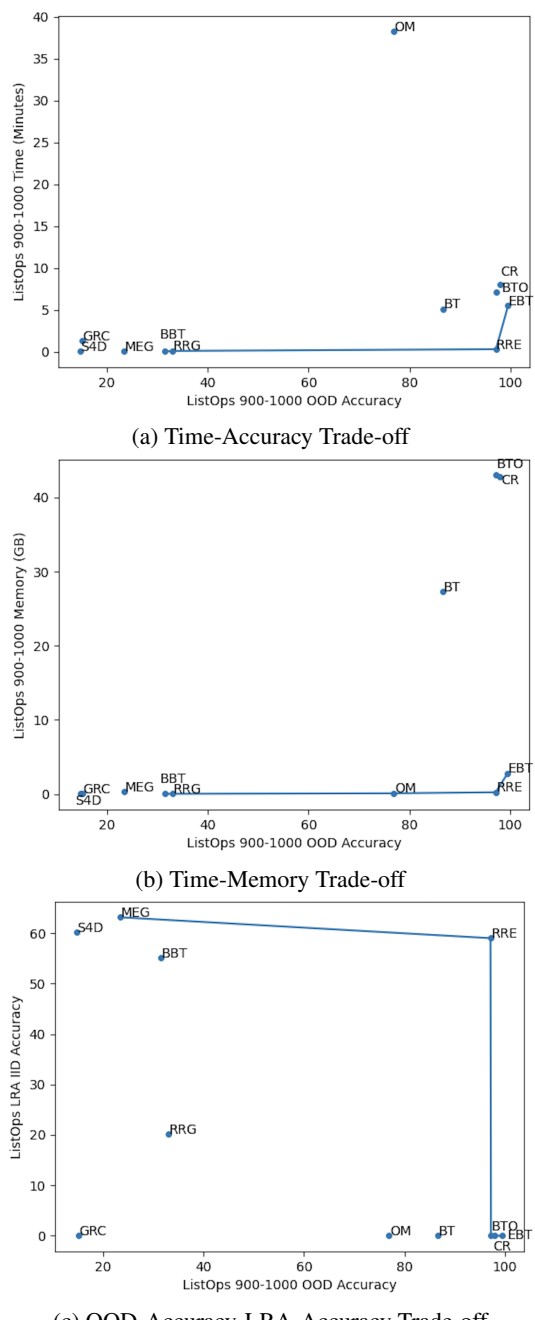

(a) Time-Accuracy Trade-off

(b) Time-Memory Trade-off

(c) OOD-Accuracy-LRA-Accuracy Trade-off

Figure 2: Scatterplot with Pareto Frontier for different trade offs. We set accuracy to 0 for LRA if the model is too expensive to practically run eg. OM, CRvNN, BT-GRC, EBT-GRC, BT-GRC OS. S4D is S4D, GRC is a recurrent model with GRC cell function, MEG is MEGA, BBT is Balanced Binary Tree GRC, RRG is RIR-GRC, OM is Ordered Memory, CR is CRvNN, BT is BT-GRC, BTO is BT-GRC OS, EBT is EBT-GRC, RRE is RIR-EBT-GRC.

Table 7: Mean accuracy and standard deviaton on SNLI [5], QQP, and MNLI [96]. Hard represents the SNLI test set from Gururangan et al. [34], Break represents the SNLI test set from Glockner et al. [27]. Count. represents the counterfactual test set from Kaushik et al. [49]. $PAWS_{QQP}$ and $PAWS_{WIKI}$ are adversarial test sets from Zhang et al. [102], MM is the mismatch dev set from MNLI, ConjNLI is the development set from Saha et al. [80], NegM, NegMM, LenM, LenMM are Negation Match, Negation Mismatch, Length Match, Length Mismatch stress test sets from Naik et al. [69] respectively. Our models were run 3 times on different seeds. Subscript represents standard deviation. As an example, $90_1 = 90 \pm 0.1$

| Models | SNLI Training | | | | QQP Training | | |
|---|---|---|---|---|---|---|---|
| | IID | Hard | Break | Count. | IID | $PAWS_{QQP}$ | $PAWS_{Wiki}$ |
| (Sequence Encoder Models) | | | | | | | |
| CRvNN | $85.3_2$ | $\mathbf{70.6_4}$ | $55.3_{17}$ | $59.8_6$ | $\mathbf{84.8_3}$ | $34.8_7$ | $46.6_6$ |
| OM | $\mathbf{85.5_2}$ | $70.6_3$ | $\mathbf{67.4_9}$ | $\mathbf{59.9_2}$ | $84.6_0$ | $38.1_7$ | $45.6_8$ |
| BT-GRC | $84.9_1$ | $70_5$ | $51_{14}$ | $59_4$ | $84.7_5$ | $36.9_{17}$ | $46.4_{12}$ |
| BT-GRC OS | $84.9_1$ | $70.3_6$ | $53.29_{10}$ | $58.6_3$ | $84.2_2$ | $37.1_8$ | $46.3_6$ |
| EBT-GRC | $84.7_4$ | $69.9_8$ | $55.6_{20}$ | $58.1_1$ | $84.3_2$ | $36.9_5$ | $\mathbf{47.5_5}$ |
| BBT-GRC | $84.4_4$ | $69.2_3$ | $65.3_8$ | $58.1_4$ | $83.6_4$ | $\mathbf{42.9_{21}}$ | $46.7_{12}$ |
| RIR-GRC | $85_{0.4}$ | $69.6_{2.9}$ | $57.2_{37}$ | $59.5_{12}$ | $84.2_{4.4}$ | $37.3_{10}$ | $45.4_{9.8}$ |
| RIR-EBT-GRC | $85.1_1$ | $70.4_7$ | $55_{29}$ | $59.6_{14}$ | $83.9_5$ | $37.2_{42}$ | $46.8_5$ |

| Models | MNLI Training | | | | | | |
|---|---|---|---|---|---|---|---|
| | Match | MM | ConjNLI | NegM | NegMM | LenM | LenMM |
| (Sequence Encoder Models) | | | | | | | |
| CRvNN | $72.2_4$ | $72.6_5$ | $41.7_{10}$ | $52.8_6$ | $53.8_{4.2}$ | $62_{44}$ | $63.3_{47}$ |
| OM | $\mathbf{72.5_3}$ | $\mathbf{73_2}$ | $41.7_4$ | $50.9_7$ | $51.7_{13}$ | $56.5_{33}$ | $57.06_{31}$ |
| BT-GRC OS | $71.7_1$ | $71.9_2$ | $41.2_9$ | $\mathbf{53.2_2}$ | $\mathbf{54.2_5}$ | $65.6_{13}$ | $66.7_9$ |
| EBT-GRC | $72.1_2$ | $72_1$ | $40.93_0$ | $52.33_{23}$ | $53.28_{22}$ | $64.92_{10}$ | $66.4_{10}$ |
| BBT-GRC | $71.1_2$ | $71.4_4$ | $40.9_{12}$ | $51.5_6$ | $52.6_8$ | $60.6_{15}$ | $61.8_{14}$ |
| RIR-GRC | $71.5_{4.8}$ | $71.6_{2.4}$ | $40.8_{9.9}$ | $50.9_{18}$ | $51.6_{18}$ | $55.2_{19}$ | $55.8_{20}$ |
| RIR-EBT-GRC | $71.8_2$ | $72.3_4$ | $\mathbf{42.4_{12}}$ | $53.1_{19}$ | $53.3_{25}$ | $\mathbf{65.7_5}$ | $\mathbf{67.3_9}$ |

# F   Pareto Frontier Analysis

We show the plots of three Pareto-frontiers focusing on three trade-offs in Figure 2. In Figure 2(a), we focus on the trade-off between ListOps-O OOD length generalization accuracy (which also correlates with Logical Inference OOD accuracy for most parts) and empirical time cost. In Figure 2(a), we focus on the trade-off between ListOps-O OOD length generalization accuracy and empirical memory cost. In Figure 2(c), we focus on the trade-off between ListOps-O OOD length generalization accuracy and ListOps LRA IID accuracy. We use the empirical time/memory cost from Table 1. In all three subfigures, RIR-EBT-GRC (RRE in the figures) remains as a Pareto-efficient solution that maintains a highly competitive trade-off. S4D, BBT-GRC, and RIR-GRC can win on the time cost and memory cost compared to RIR-EBT-GRC, but with a sharp degradation of OOD performance in structure-sensitive tasks (logical inference, ListOps). While OM, CRvNN, BT-GRC, BT-GRC OS, and EBT-GRC can outperform RIR-EBT-GRC (to an extent) on OOD length generalization accuracy in ListOps and logical inference, they come with much more exorbitant time/memory cost.

# G   Extended Related Works

**RIR-related Approaches:** Many approaches have used chunking for hierarchical processing [90, 98, 59, 81, 15, 79, 36, 103, 97, 83, 72, 10, 9, 43, 60] to efficiently tackle long-range inputs. However, they generally neither go all the way in using a full balanced tree recursion (with dynamic layer depth) nor concentrate on remaining competitive on structure-sensitive tasks. One exception is Sliced RNN [101] which introduces a model similar to RIR-GRC but with an older recurrent cell. Some

Table 8: Accuracy on LRA [92]. The results in the top block are copied from the respective papers. For our models, we report the mean of 2 runs. Subscript represents standard deviation. As an example, $90_1 = 90 \pm 0.1$

| Model | LRA | | |
| | ListOps | Text | Retrieval |
| --- | --- | --- | --- |
| Transformer [60] | 37.11 | 65.21 | 79.14 |
| Local Attention [92] | 15.82 | 52.98 | 53.39 |
| Linear Transformer [92] | 16.13 | 65.90 | 53.09 |
| Reformer [92] | 37.27 | 56.10 | 53.40 |
| Sparse Transformer [92] | 17.07 | 63.58 | 59.59 |
| Skinhorn Transformer [92] | 33.67 | 61.20 | 53.83 |
| BigBird [92] | 36.05 | 64.02 | 59.29 |
| Performer [92] | 18.01 | 65.40 | 53.82 |
| Linformer [92] | 35.70 | 53.94 | 52.27 |
| Longformer [92] | 35.63 | 62.85 | 56.89 |
| Transformer TLB [18] | 37.05 | 81.88 | 76.91 |
| H-Transformer-1D [106] | 49.53 | 78.69 | 63.99 |
| S4-LegS [32] | $59.6_{0.7}$ | $86.82_{1.3}$ | $90.90_{1.5}$ |
| S4D-Inv [30] | $60.18_{3.5}$ | $87.34_2$ | $91.09_{.1}$ |
| SGConv [55] | 61.45 | 89.20 | 91.11 |
| ChordMixer [50] | 60.12 | 88.82 | 90.17 |
| S5 [86] | 62.15 | 89.31 | **91.40** |
| Liquid S4 [37] | 62.75 | 89.02 | 91.20 |
| Long Conv, Exp + Squash [24] | 62.2 | 89.6 | 91.3 |
| MEGA [60] | **63.14** | **90.43** | 91.25 |
| LRU [73] | $60.2_8$ | $89.4_1$ | $89.9_1$ |
| S4D-Inv (reproduction) | $59.3_0$ | $88.18_{0.4}$ | $91.17_{.6}$ |
| BBT-GRC | $55.15_1$ | $85.69_{0.7}$ | $90.78_{1.6}$ |
| RIR-GRC | $20.18_{5.7}$ | $86.894_{1.3}$ | $88.24_{10}$ |
| RIR-EBT-GRC | $59.0_4$ | $87.97_{1.6}$ | $88.99_{5.2}$ |
| -S4D | $59.53_1$ | $88.13_{1.3}$ | $89.74_{3.9}$ |

approaches take a block-recurrent [14, 44, 46, 18, 8] approach where some $k$-sized chunk is used in a recurrent setup wherein each chunk is processed by a Transformer or some other neural module. These approaches are similar to the RIR framework where the outer recursion is made of some $k$-ary chain-tree structured recurrence (instead of balanced-tree recursion), and the inner 'recursion' is implemented for example by a Transformer. A concurrent work Chi et al. [11] also uses a similar structure as RIR but with a Transformer stack replacing the inner inner RvNN.

**RvNN:** Pollack [76], Goller and Kuchler [29] first introduced Recursive Neural Networks (RvNNs) for bottom-up representation build up based on directed acyclic graphs or trees. Socher et al. [87, 88, 89] extended RvNNs for NLP, and [104, 91, 53, 105] adapted RvNNs with LSTM cells [40]. Le and Zuidema [54], Maillard et al. [64] proposed a chart-based method for simulating bottom-up RvNNs based on CYK algorithm. Drozdov et al. [19, 20], Hu et al. [41, 42] built upon it. Shi et al. [84], Munkhdalai and Yu [68] explored heuristics-tree-based RvNNs including balanced trees. Multiple works explored memory-augmented RNNs for simulating RvNNs [7, 100, 63, 82, 21, 22]. Arabshahi et al. [1] proposed a technique for augmenting recursive cells within RvNNs with a stack-like memory mechanism.

There are other related models [65] that explore structure-sensitive tasks like mathematical equation verification.

# H  RIR for Language Modeling

Language modeling is today crucial for pre-training and scaling models for competitive performance in several realistic tasks. Nevertheless, directly using RIR-EBT-GRC for language modeling is not straightforward. We discuss some speculative directions we can go forward in re-framing RIR-EBT-GRC for language modeling.

**Masked Language Modelingr:** In a concurrent work, we propose a method to create contextualized token representations for EBT-GRC or BT-GRC [78] through parent attention (making terminal nodes attend to its direct and indirect parent nodes based on the recursively built-up latent tree). The same principles can be used for RIR-EBT-GRC as well. After that RIR-EBT-GRC can be used for masked language modeling as standard model setups like BERT [17]. It can be also stacked with other types of models like Transformers if needed. Such a setup can also be used as an encoder in a Seq2Seq (Encoder-Decoder) architecture which can be pre-trained with relevant language modeling policies.

**Sentence Encoding in a Loop:** There is a simple method for using any sentence encoder (including RIR-EBT-GRC) for autoregressive language modeling. At any timestep $t$, we can take the sequence of past tokens $s_{1:t-1}$ as the input and get a sentence encoding - say, $h_t$ (representing the compression of $s_{1:t-1}$). Then we can predict the next token from $h_t$ as $s_t$ by projecting $h_t$ into a probability distribution over a vocabulary. However, this is a very costly method, because we have to run the whole recursion over the whole generated sequence in every timestep. It would lack the memory reuse capacities of pure recurrent models (using chain-like tree structures), and it would lack the parallelism of Transformers trained with teacher-forcing. Nevertheless, it is a method that can be used given enough resources.

**Block-Recurrent Sentence Encoding in a Loop:** One way to address the cost of the above approach is to instead take a block-recurrent approach [44]. Combined with RIR-EBT-GRC, this will be similar to taking a three-level nested recursion where the outermost recursion is recurrence and RIR-EBT-GRC will be the inner two-level recursion. In this case, in a timestep RIR-EBT-GRC receives as input not the whole past sequence, but part of the past sequence with the earlier parts recurrently compressed into some finite memory blocks from the outermost recurrence. Thus, the inner recursions (RIR-EBT-GRC) with the "sentence encoding in loop" setup can run with a bounded input size. There could be other creative ways to reduce the overall cost. For example, we can attempt to run the sentence-encoder loop periodically after some intervals intermixing fast (for example recurrently generating from the last sentence encoding instead of re-encoding the whole sequence) and slow generation (re-encoding the whole input sequence).

**Parameter Scaling:** To be a competitive language model, we need some ways to scale the parameter size of RIR-EBT-GRC. This is challenging for recursive models. The challenge is that the simplest way to increase the parameters is by increasing the hidden state sizes or overall increasing the computation of the recursive cell. Particularly, the increase has to be significant if we want to compete with parameters close to multi-layered models. However, because the recursive cell has to be repeatedly applied for an arbitrary number of times, the overall computation becomes very expensive. Similar challenges also occur for Universal Transformer [71]. One way out of this problem is to stack more layers of RIR-EBT-GRC or alternate models like Transformers or SSMs after creating token-contextualized representations using parent attention [78]. This can also lead to the utilization of a mixture of inductive biases. We can then keep the parameters in the recursive cells themselves to a more manageable level. Another direction to look for can be a modular or mixture-of-expert-based approach. This can allow increasing parameters while keeping computation more manageable via sparse activation. For example, we can attempt to recursively select and activate only a sparse region of parameters (some sparsely selected modules), to keep computation expense controlled while increasing the total parameter budget.

Even after all these, the overall setup would probably not be as fast as Transformer training. But speed or even mere surface capacity to handle big contexts is not everything - and can come at other costs [56, 58].

# I  Beam Alignment Intuition

In Figure 3, we present a visualization of the String-it solution and in Figure 4, present a visualization of the Beam Alignment solution. In the visualizations, we consider a simple scenario where $\lfloor \frac{n}{k} \rfloor =$

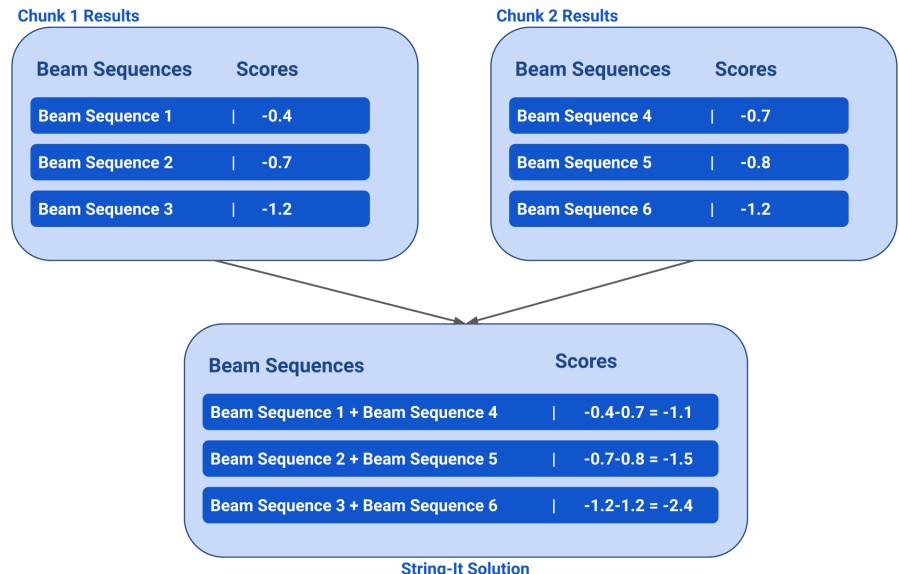

Figure 3: Visualization of String-it Solution. Beam Sequence x + Beam Sequence y indicates concatenation.

$2, b = 3$ ($n =$ input sequence length for the recursion, $k =$ chunk size, $b =$ beam size). The String-it solution (Figure 3) is to simply concatenate the beam sequences (and add the corresponding scores) from all lists that occur in the same position. For example, in Figure 3, we concatenate the first beam (Beam Sequence 1) from the first list (chunk 1 results) with the first beam (Beam Sequence 4) from the second list (chunk 2 results). Similarly, we concatenate the second beam from the first list with the second beam from the second list, and so on.

However, we want to create "high-scoring" beam combinations with more probability. The String-it solution does not care for that. For example, "Beam Sequence 1 + Beam Sequence 5" would be the second highest-scoring beam combination, but it cannot be ever selected by String-it since Beam Sequence 1 is in the first position of the first list and Beam Sequence 5 is in the second position of the second list - that is, they are not in the same position.

Thus, we propose Beam Alignment where we take a stochastic approach towards the ideal of biasing the preservation of high-scoring combinations. For this approach, instead of immediately applying the String-it solution, we make the beams in each list stochastically 'compete' for their positions. Essentially, we want high-scoring beams to be more likely to occupy more positions in the beam lists from each chunk. This is done by simply sampling a beam per list position (for each of the $b$ positions) according to the normalized beam score distribution. The result is that the beam lists will be now filled with mostly the higher-scoring beams (See the results after 'Sample' in Figure 4). Next, if we simply apply the String-it solution at this point, it automatically leads to high-scoring combinations, as we wanted, because of the prior sampling. Now there is a possibility of combinations like "beam sequence 1 + beam sequence 5" to arise because the sampling step allows the same beam to occupy multiple positions.

Overall, as we can see, in the simulation of Beam Alignment (Figure 4) the resulting combined beams tend to have higher scores than in the case of the direct application of String-it (Figure 3).

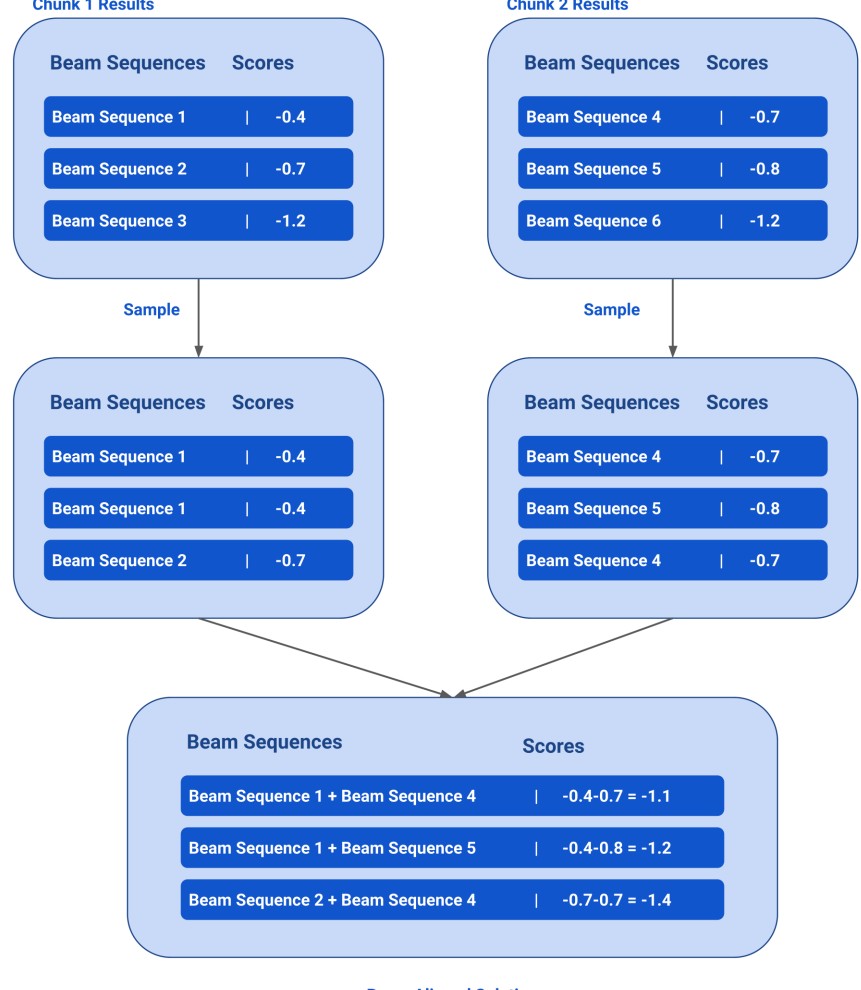

Figure 4: Visualization of Beam-Align Solution. Beam Sequence x + Beam Sequence y indicates concatenation.

## J   Architecture details

**Gated Recursive Cell (GRC):** For the Recursive Cell that is the $R$ function (used within the inner RvNN), we use the GELU-variant of Gated Recursive Cell (GRC) [13] (originally a ReLU variant was proposed in [82]) unless mentioned otherwise. The cell function is formalized below:

$$\begin{bmatrix} l, \\ r, \\ g, \\ h \end{bmatrix} = \text{GeLU}\left(\begin{bmatrix} child_l; \\ child_r \end{bmatrix} W_1^{rec} + b_1\right) W_2^{rec} + b_2 \tag{10}$$

$$p = LN(\sigma(l) \odot child_l + \sigma(r) \odot child_r + \sigma(g) \odot h) \tag{11}$$

Here - $\sigma$ is $sigmoid$; $[;]$ represents concatenation; $p$ is the parent node representation built from the children $child_l$ and $child_r$, $W_1^{rec} \in I\!\!R^{2 \cdot d \times d_{cell}}$; $b_1 \in I\!\!R^{d_{cell}}$; $W_2^{rec} \in I\!\!R^{d_{cell} \times 4 \cdot d}$; $b_2 \in I\!\!R^d$, and $l, r, g, h \in I\!\!R^d$. $LN$ is layer normalization. $d_{cell}$ is generally set as $4 \times d$.

**Scorer:** Similar to [78], for the $scorer$ we used a two layer MLP as:

$$e_i = scorer(x, y) = W_2^s \text{GeLU}(W_1^s[x; y] + b_1^s) + b_2^s \tag{12}$$

Here, $W_1^s \in I\!\!R^{d_s \times 2 \cdot d_s}, W_1^s \in I\!\!R^{1 \times d_s}, b_1^s \in I\!\!R^{d_s}, b_2^s \in I\!\!R$.

**Sentence Encoder Models:** For sentence encoding, the architectural framework we use is the same Ray Chowdhury and Caragea [77]. We use a Siamese dual-encoder setup for sentence-pair tasks as prior works [77].

**S4D Pre-Chunk Layer:** For the RIR-based models, before starting the RIR-loop, we use a bidirectional S4D [30] layer for pre-chunk processing. Let us represent the basic single layer State space component (without non-linearity) of S4D as $\text{S4D}_f : I\!\!R^{n \times d} \to I\!\!R^{n \times d}$ (forward S4D) and $\text{S4D}_b : I\!\!R^{n \times d} \to I\!\!R^{n \times d}$ (backward S4D). We express the overall pre-chunk processing layer using S4D below:

$$x_f = \text{GELU}(\text{S4D}_f(x)) \tag{13}$$

$$x_b = \text{GELU}(\text{S4D}_b(x.\text{flip}())).\text{flip}() \tag{14}$$

$$x_{cat} = [x_f; x_b] \tag{15}$$

$$out = \text{sigmoid}(x_{cat}W_u) \odot (x_{cat}W_v) + x \tag{16}$$

$$inp = LN(out W_{init}) \tag{17}$$

Here $W_u, W_v \in I\!\!R^{2d \times d}, W_{init} \in I\!\!R^{d \times d}$, and LN is Layer Normalization. Eqn. 16 is the application of gated linear unit [16] and a residual like Gu et al. [30] for feature mixing, and Eqn. 17 is the initial leaf transformation layer that is used in prior works [82, 13, 77].

**S4D:** When using pure (multi-layered S4D without RIR-models) S4D models, we use similar settings as Gu et al. [30]. We found it more beneficial to use the fused kernel for bidirectionality within their codebase[8] when using S4D (setting 'bidirectional=True' in their S4 function argument).

## K   Hyperparameter details

For prior sentence encoder models, we use the same hyperparameters as [77] for all the datasets. $d_s$ for EBT-GRC is set as $64$. For RIR-EBT-GRC, RIR-inference was only used for Text LRA and Retrieval LRA for computational efficiency. We always use RIR-inference for RIR-GRC because we found to be better by validation accuracy. We use chunk size as 30 for all RIR-models in all tasks, except Retrieval LRA where we set chunk size as 5 for computational efficiency (otherwise it gets slow). For RIR-EBT-GRC, we use a beam size of 7 for all tasks except Retrieval LRA where we use a beam size of 5. Every other hyperparameters are unchanged from BT-GRC for RIR-models

---

[8]`https://github.com/HazyResearch/state-spaces`

or BBT-GRC for the earlier tasks. For the new tasks, we share the hyperparameters for MNLI with that of SNLI, QQP, and Retrieval LRA (except that for Retrieval LRA we use a smaller hidden state size - 128 for parameter control and no pre-trained embeddings). We share the hyperparameters for IMDB and Text LRA (except that for Text LRA, we use smaller hidden state size - 128; again, for parameter control, and no pre-trained embedding). We share the hyperparameters of ListOps-O, ListOpsMix, Logical Inference, and ListOps LRA for all models. We initialize S4D (whether when using the pure S4D model or S4D for pre-chunk processing) in S4D-Inv mode and use billinear discretization. For LRA tasks, we use the same hyperparameters as Gu et al. [30] for S4D. We use the same hyperparameters as used for ListOps LRA Ma et al. [60] for MEGA as run in ListOps-O and Logical Inference. Note that all the natural language tasks (except LRA-related ones) are trained with frozen 840B Glove Embeddings [75] same as Ray Chowdhury and Caragea [77]. All models were trained on a single Nvidia RTX A6000.

