# OpenReview forum: "Recursion in Recursion: Two-Level Nested Recursion for Length Generalization with Scalability"
_NeurIPS.cc/2023/Conference — NeurIPS 2023 poster_

### Official Review · Reviewer_3jTN · 2023-07-10

**Soundness:** 3 good
**Presentation:** 2 fair
**Contribution:** 2 fair
**Rating:** 4
**Confidence:** 3

**Summary:**

Recursion in Recursion combines the efficiency characteristics of Binary Balanced Tree Recursive Neural Networks (BB-Tree RvNNs) and quality characteristics of Beam Tree RvNNs by applying the recursion in a hierarchical fashion.

Doing so is not straightforward and requires several "modifiers" such as Beam Alignment and being careful about chunk preprocessing in the outer recursion.

Table 1 demonstrates that the method is efficient in both time and memory. Table 2 demonstrates that the model quality is compromised but not by much.

**Strengths:**

The paper provides necessary background which I appreciated since I am not too familiar with RvNNs. Experiments seem well done and reproducible. The method is efficient as promised and still maintains the ability to generalize to sequence length.

While the method is still slow and seem less mature than other methods like transformers, I think RvNNs are interesting and worthy of discussion because of their explainability and ability to generalize to longer sequences.

**Weaknesses:**

While interesting, the method seems rather complex to apply generally with 2 levels of recursion and necessary modifiers to integrate them effectively.

The beam alignment section I found particular confusing. Maybe there could be a figure to help understanding?

The inability to generalize on argument length could use more discussion.

**Questions:**

I would be curious about how frequently bad chunk processing happens and the model's ability to recover? How much does S4D help here?

**Limitations:**

The authors have a limitations section that discuss slowness and lack and difficulty to generalize.

To me the the method seems complex. I am not sure if it is broad applicable given that it is still slow and model quality is still lacking.

---

> ### Author Rebuttal · Authors · 2023-08-10
>
> Thank you for the review.
>
> **Re Weaknesses**
>
> 1. Please see the general response for the clarification on the complexity.
>
> 2. We provide figures and additional intuition on beam alignment  in the general response.
>
> 3. Argument generalization was a recently proposed challenge, and currently, most models struggle with it - including GoldTreeGRC (a model with ground truth structure data). We will need to investigate the cell function or other training strategies for proper argument generalization which would require separate work and is left as future work. Note that while it’s a limitation of our model in not being as good at argument generalization, this limitation, at the moment, is nearly universal - shared by most of the popular models - S4D, Transformers (Transformers struggle even IID [2]), MEGA, and others. Only one or two models lacking explicit structure-information (OM, EBT-GRC) are moderately good but still falling behind 90%.
>
> **Re Questions**
>
> 1. Bad chunks should happen nearly 100% of the time since ListOps is not set up to have balanced structures at any level. We show the comparison of “RIR-EBT-GRC” vs “RIR-EBT-GRC-S4D” in various Tables (please see Table 2 and Table 4). As we find there, S4D actually does not help much. So this is an area open for improvement/further investigation. The ability to “recover” the right structure through bad chunks is hard to check because if it recovers it has to do it internally by organizing information in its hidden states in the right way (analogous to how an RNN may model tree structures [3]) which is more of a blackbox process.
>
> **Re Limitations**
>
> 1. Note that the difficulty to generalize (e.g., in argument generalization) is also present in other existing models like S4D, MEGA, and Transformers, which fail in length generalization too [1,2]. RIR-EBT-GRC does much better than them.
>
> 2. While RIR-EBT-GRC is slower than S4D/MEGA/BBT-GRC - their speed comes at the cost of worse performance in structure-sensitive contexts. Moreover, our proposal - RIR-EBT-GRC is several times faster than other powerful RvNN models (OM, CRvNN, BT-GRC) - making a massive computational improvement for RvNN models (Table 1) while retaining decent performance (Table 2, Table 3). For example, it’s 100s of times faster than OM.
>
> Also please see the Pareto frontier graphs in the general response pdf which show how RIR-EBT-GRC gains a better trade-off than others.
>
> [1] The Importance of Being Recurrent for Modeling Hierarchical Structure - Tran et al. EMNLP 2018
>
> [2] Ordered Memory - Shen et al. Neurips 2019
>
> [3] Tree-Structured Composition in Neural Networks without Tree-Structured Architectures - Bowman et al. International Conference on Cognitive Computation: Integrating Neural and Symbolic Approaches 2015

---

### Official Review · Reviewer_CUBJ · 2023-07-13

**Soundness:** 3 good
**Presentation:** 3 good
**Contribution:** 3 good
**Rating:** 7
**Confidence:** 3

**Summary:**

The paper studies a “recursion in recursion” (RIR) approach, the outer loop being a balanced k-tree recursive NN, and the inner loop a general recursive NN that is based on a beam tree. The goal is to obtain O(k*log_k(N)) time complexity. O(2*log_2(N)) is a special case of k=2 (for binary trees) and O(N) is the special case of k=N for RNNs. The authors study (mostly) the tradeoff between performance on ListOps and Long Range Arena (LRA). Namely, RIR seems competitive to strong baselines on LRA (e.g. state space models), yet it generalizes better on longer lengths in ListOps.

**Strengths:**

S1. The paper is very well written. The research process is embedded in the structure of the paper: “motivation -> problem -> solution -> new problem -> new solution -> etc.”

S2. The  modifications are quite reasonable and the experiments are thorough (the appendix was helpful too).

S3. The papers’ evidence matches the claims quite well and the limitations are well-acknowledged.

**Weaknesses:**

W1. It is very hard to read through the tables. Since you are studying a trade-off of time/ memory efficiency and performance on ListOps, and likewise ListOps vs. LRA performances, I would expect scatter plots that trace out a Pareto frontier and where your RIR contributions lie on this frontier.

W2. Despite the work being very engineeringly sound and well-presented, one could argue that the contributions as “combinatorial” and mostly heuristic vs. fundamental. That is OK, but one could consider it a weakness within the context of NeurIPS.

W3. Following up on 2., it may be good to discuss the significance of your contribution to larger tasks and broader / realistic data distributions. For example, how would RIR might be a part of a state-of-the-art LLM architecture in the future?

W4. Perhaps the most exciting discussion for me is about different inference strategies (whether to use RIR or not). I acknowledge that you have addressed some of that in the SM, but I would have expected to see a bit more discussion in the main text.

**Questions:**

Q1. Why do you think that RIR is not very helpful for inference? I like it because of the more efficient inference that it enables. Is it possible to demonstrate the efficiency/ accuracy trade-off with another Pareto frontier plot?

**Limitations:**

the authors adequately addressed the limitations

---

> ### Author Rebuttal · Authors · 2023-08-10
>
> Thank you for the review.
>
> **W1:** We will add the Pareto frontier graphs. We currently show them in the General Response PDF.
>
> -----
>
> **W3:** At this point, we can suggest some speculative moves we can make to set up RIR-EBT-GRC in an LLM context.
>
> First, we need to think about parameter scaling. The naive way to scale parameters would be simply increasing the dimensions of the recursive cell. This is inefficient in recursive contexts (this would lead to repeating a large layer for indefinite times based on sequence length). Some alternatives can be to look for some sort of modular/MoE approach (where each recursion operates on some sparse parameters from a bigger parameter set) or to try some form of stacking (just as we, sometimes, stack RNNs). Now, stacking RIR-EBT-GRC is not immediately possible because it’s a sentence-encoder (it compresses a sequence of vectors into a single vector after which there is not much room to add another sequence model). However, one thing we can do is try to create a token contextualization by sending top-down signals from representations of whole to representations of parts (initial token representations) (similar to [1]). In fact, we are exploring this idea in a concurrent paper by using a parent attention mechanism. This can allow stacking the output of RIR-EBT-GRC with other RIR-EBT-GRCs, Transformers, SSMs, or CNNs (and the overall model can utilize a mixture of inductive biases).
>
> Second, after parameter scaling, another main difficulty is utilizing it in a causal LLM training context. A difficulty is that RIR-EBT-GRC is a bidirectional model. We can still train a bidirectional model for LLM by sequentially entering the expanding context in a loop but this would be slow. To make it tractable, first, we can use a Block-Recursive style framework [2] and second, we can limit the use of RIR-EBT-GRC for specific intervals (use it sparsely), and use a simpler model that use the latest RIR-EBT-GRC-created past hidden states + free tokens (causally masked) to make predictions in between. Some loosely related ideas are used in [5] (they are also trying to work with looping a bidirectional model for autoregressive generation).
>
> Even after all these, the overall setup would probably not be as fast as Transformer training. But speed or even mere surface capacity to handle big contexts is not everything - and can come at other costs [3,4].
>
> [1] Head-Lexicalized Bidirectional Tree LSTMs - Teng et al. TACL 2017
>
> [2] Block-Recurrent Transformers - Hutchins et al. Neurips 2022
>
> [3] Exposing Attention Glitches with Flip-Flop Language Modeling - Liu et al. ArXiv 2023
>
> [4] Lost in the Middle: How Language Models Use Long Contexts - Lie et al. ArXiv 2023
>
> [5] Real-World Compositional Generalization with Disentangled Sequence-to-Sequence Learning - Zheng et al. ArXiv 2022
>
> -----
>
> **W4-Q1:** We will add RIR inference details in the main text. We will also add trade-off information. But overall, to answer the question - in most tasks (including LRA) RIR-inference seems to work nearly as well as non-RIR-inference. So in most cases, RIR inference is a “win-win” (several times faster + nearly the same accuracy). However, when a model is working in a structure-sensitive context, it may find it difficult to learn to length generalize as well through an RIR format (it still does better than S4D/MEGA in the ablation Table 1 in Appendix) -- because ultimately the balanced-tree enforcement is a restriction on the search space that is generally unsuited for structure-sensitive tasks (and merely done for efficiency). However, the existence of data length <= chunk size, leads to some non-RIR training instances which probably helps teach the inner recursion to learn to length-generalize. Thus, testing without RIR inference (i.e., running the inner recursion model in the full input) can still lead to length generalization and be even better (in structure-sensitive contexts when the inner recursion indeed learns to utilize the relevant structures in a length-generalizable manner) due to the lack of balanced tree enforcement work. In these cases, RIR inference can reduce the performance compared to non-RIR inference. Note, however, since the use of -Beam Align or +Random Align during training reduces the non-RIR inference accuracy (these changes only make an impact for instances running in RIR mode), the RIR instances (i.e., instances with sequence length > chunk size) should be still providing some training signals  in helping the inner recursion model to learn to generalize.

---

> > ### Comment · Reviewer_CUBJ · 2023-08-19
> > **I read the rebuttal. I appreciate the clarifications and additions and will raise my score.**
> >
> > Thanks. I would encourage you to add the discussions in the main text/ SM.

---

> > > ### Author Response · Authors · 2023-08-19
> > >
> > > Thank you for raising the score. We will add the discussions.

---

### Official Review · Reviewer_yLki · 2023-07-17

**Soundness:** 2 fair
**Presentation:** 2 fair
**Contribution:** 2 fair
**Rating:** 4
**Confidence:** 2

**Summary:**

The authors propose a new tree-style RNN that combines the computational efficiency of tree RNNs with the computational power of more complex tree RNNs. The method makes each node perform a recursive computation, so there are logarithmically many nodes but they are more expressive than the standard tree network. The resulting model can operate on long sequences comparably to state space models and much better than vanilla transformers.

**Ethics Review**: It appears to me that the authors broke the integrity of the double-blind reviewing procedure. In lines 144-146, the authors write: "we use an efficient variant of BT-RvNN (EBT-RvNN). We propose and explore EBT-RvNN in a concurrent work". NeurIPS policy states: "If you need to cite one of your own papers, you should do so with adequate anonymization to preserve double-blind reviewing.  For instance, write “In the previous work of Smith et al. [1]…” rather than “In our previous work [1]...”). If you need to cite one of your own papers that is in submission to NeurIPS and not available as a non-anonymous preprint, then include a copy of the cited anonymized submission in the supplementary material and write “Anonymous et al. [1] concurrently show...”).". As such, I am flagging it for an ethics review.

**Edit after rebuttal**: I have increased my score from 3 to 4.

**Strengths:**

The paper improves on other tree-based RNNs by reducing the computation.

**Weaknesses:**

This may be my lack of understanding on this subject area, but I think this is an overly complicated method that does not outperform existing methods or provide a clear benefit over them. There is some value to advancing a subfield of ML (i.e., tree-based RNNs), but I find it difficult to see the significance of this work. It may also be because of issues with the writing, see below.

Besides the complexity of the method and the performance issues, the authors also do not benchmark on many tasks. They focus on the ListOps task from the Long Range Arena benchmark and don't consider Mega, which has achieved SoTA on this task.

**Writing**: Overall the writing is very difficult for someone who has not been actively thinking about this problem and working with tree-style RNNs. I found it very difficult to read the introduction and abstract because there is too much jargon. Below are some suggestions and questions that may help the authors improve their writing for an audience that is not working on tree-based RNNs. I generally recommend refactoring the writing into formal definitions so it is clear to see how the inner and outer recursions work together. As it stands, I couldn't even write down how your method would actually process a given input.
- Can you make a figure illustrating these different trees instead of trying to describe them in the text?
- I don't know what a "strong RvNN" is or what it means to process only some arguments that are sent in from an outer loop. (lines 75-79)
- Can you introduce clearly what ListOps is and why it is important to be length-generalizable for that task?
- There is not enough information about these other methods in the tables and how they work.

**Questions:**

1. In what situations would I use this method instead of other stronger models (e.g., Mega)?
2. Is there any benefit to using a tree-based model instead of a modified transformer architecture?

**Limitations:**

The authors discuss the limitations clearly, which I appreciate. They mention that their architecture improves on tree-based RNNs but is much slower than other, seemingly stronger models. It also seems to not be able to learn from long sequences, which may be the point of a long-context model. One thing they do not mention is that their architecture seems to be very complicated and has many tricks involved. They also only evaluate on a few tasks, and they seem to be very focused on ListOps in particular. For other tasks, their method performs comparably to existing tree-based models (lines 325-326).

---

> ### Author Rebuttal · Authors · 2023-08-10
>
> Thank you for the review.
>
> **Re Ethics:**
>
> We will not say too much about this given Ethics Reviewers have already checked it out. We didn’t provide any deanonymizing information about the work. While we haven’t cited the work or provided the anonymized copy at the moment (we will cite it in the final copy), we have provided all the necessary details pertinent to this work in the main paper and the appendix. If there are still some questions, we will be happy to clarify.
>
> **Re Weaknesses:**
>
> * Please see the general response comment on complexity.
>
>
> * Significance: While models like  S4D, and MEGA are generally powerful, they can struggle in structure-sensitive tasks and in length generalization (eg. in Logical Inference, and ListOps). On the other hand, models like OM, CRvNN, BT-GRC, EBT-GRC can generalize more robustly but they are very slow to run (please see Table 1). Our work proposes a framework to strike a better balance between length generalization capacity and competence in long sequence tasks (as alluded in the title of the paper). Also see the Pareto Frontier graphs in the General Response PDF for some visualizations. Moreover, as you noted, there is value in exploring niche areas and alternative approaches too (so as to not put all the eggs in one or two baskets) even if right now there are limitations to RvNNs.
>
>
> * We also have results of logical inference (Table 3), results on IMDB, AAN retrieval from LRA (Table 4), and other NLP tasks in the Appendix. We added MEGA results in the general response pdf.
> We will improve the writing.
>
>
> * “strong RvNNs” refers to the modern RvNNs that can perform well in tasks like ListOps/Logical Inference (eg. OM, CRvNN, BT-GRC).
>
>
> * ListOps is an arithmetic task with nested list operators example: “[MAX [MED [MED 1 [SM 3 1 3 ] 9 ] 6 ] 5 ]” the answer is 6 (here MED is median, and SM is modulo 10 summation). We will add more details in the paper.
>
>
> * Importance of length generalization: first, we can ask ourselves how do we know if a neural network learns the “right” function and is not simply exploiting some surface statistical bias from the training dataset. One simple way to test this is to test the model in unseen datasets (the test set). But this is not necessarily ideal. Test sets, if it is from the same (IID) distribution as the training set, may share some statistical biases which the model can be exploiting. So what can we do? One thing we can do is to think of specific isolated factors of generalizations that a neural network should exhibit if it learns the desired function from the data. For example, we may want the neural network to be robust against local perturbations. So we may try to test in some adversarially perturbed datasets. Or we may want the neural network to be robust to counterfactual modifications - and we may then create counterfactual test sets [1]. Similarly, “length generalization” is one such factor of generalization to test a model on. Particularly, if a human is taught arithmetic on examples up to length $100$, we would expect them to be able to solve arithmetic problems on length $>100$ without ever seeing higher-length examples before. We can check if a Neural network can exhibit similar robustness/generalization capacities. Overall, length generalization has been a topic of interest for a long time in the ML community.
>
>
> * Regarding details on other models, OM [2] is a form of stack-augmented RNN (think of pushdown automata), whereas CRvNN [3] is like a standard RvNN but it softens the sibling selection and updates mechanisms. We will add more details on these models.
>
>
>
> **Re Questions:**
>
> 1. You could use our method instead of Mega when you want better OOD generalizations, particularly with tasks that require stronger structural bias.
>
> 2. Modified Transformer and Tree-based models are not a dichotomy. There are Tree-based Transformers too. CRvNN is also Transformer-like in some respects. Regardless, models that are closer to the structure of Transformers have generally failed to show robust length generalization in tasks like ListOps, mathematical tasks, and propositional logical inference [2,4]. Neural Data Router [5] makes good progress but it still shows limitations in generalizing to order of magnitude higher depths and lengths (as studied in the “Beam Tree Recursive Cells” paper in the appendix).
>
>
> [1] Learning The Difference That Makes A Difference With Counterfactually-Augmented Data - Kaushik et al. ICLR 2020
>
> [2]  Ordered Memory - Shen et al. Neurips 2019
>
> [3]  Modeling Hierarchical Structures with Continuous Recursive Neural Networks - Ray Chowdhury et al. ICML 2021
>
> [4] The Importance of Being Recurrent for Modeling Hierarchical Structure - Tran et al. EMNLP 2018
>
> [5] The Neural Data Router: Adaptive Control Flow in Transformers Improves Systematic Generalization - Csordas et al. ICLR 2022

---

> > ### Comment · Reviewer_yLki · 2023-08-11
> >
> > Thanks to the authors for their response. RE: the ethics review, I just wanted to do my due diligence following the instructions.
> >
> > The authors did a good job of addressing my concerns, particularly with respect to Mega and other Transformer-based models for long sequence modeling. I think the method is sound, and the authors' claims seem reasonable. Ultimately, I still have my concerns about the significance of the work, given how slow and complicated the method is. I recognize the authors' efforts to alleviate concerns about the difficulty of implementing this method, but I still feel that it is a much more complex method for a small gain.
> >
> > Ultimately, I am raising my score 3 -> 4 because I think the work is sound and represents an improvement over other works. I will leave it to the AC and SAC to judge the significance of this work (i.e., the utility of the proposed method) in the subfield of long-range sequence modeling.

---

> > > ### Author Response · Authors · 2023-08-11
> > > **Clarification on the gain**
> > >
> > > Thank you very much for raising the scores.
> > >
> > > Re:
> > >
> > > > I still feel that it is a much more complex method for a small gain.
> > >
> > > We acknowledge (as we did in the limitations of the main paper) the limitation that RIR-EBT-GRC is slower compared to BBT-GRC, S4D, and MEGA but we would like to quantitatively highlight the gains:
> > >
> > > * RIR-EBT-GRC reduces the training time of EBT-GRC (in Table 1's 1500-2000 sequence length settings) from $10.5$ minutes to $0.3$ minutes. The gain is $35$x speed up here. RIR-EBT-GRC also reduces $10.97$ GB memory of EBT-GRC to $0.43$ GB in the same settings which is ~$25$ times memory reduction while maintaining comparable performance (as you acknowledged: "their method performs comparably to existing tree-based models"). The overall speed-up compared to OM ($255$x) and memory reduction compared to CRvNN/BT-GRC OS ($188$x) is even greater.
> > >
> > > * Moreover, MEGA gets $23.5$% accuracy in 900-1000 OOD settings of ListOps. RIR-EBT-GRC gets $97.1$%. That's a substantial difference of $73.6$%.
> > >
> > > * MEGA gets ~$64$% in logical inference with 12 operators (after training on <=6 operators). RIR-EBT-GRC gets ~$93$%. That's nearly a $29$% difference.
> > >
> > > We acknowledge that in IID settings for long-range sequence modeling, this method is not ideal compared to MEGA/S4D and others.  However, the significance motivated for the proposal was not "doing better than others in long-range settings", but **striking a balance** between scalability and length generalizability (including in short-range settings) while maintaining reasonable performance in long-range settings.
> > >
> > > There are also strategies that follow from the results of the paper to mitigate the speed limitations:
> > >
> > > * By reducing the chunk size $k$ we can get RIR-EBT-GRC closer to BBT-GRC (a special case of RIR-EBT-GRC when k=2) and start to approach the speed of S4D (in Table 1 in the main paper BBT-GRC is as fast or faster than S4D). Using small $k$ is still competitive in LRA text sets (eg. BBT-GRC performance in Table 1). Although smaller $k$ provides lower performance in OOD settings of ListOps, it's still much better than MEGA/S4D (see k (chunk size)$=10$,k (chunk size) $=20$ results in Ablations Table 1 in Appendix) in comparable settings.
> > >
> > > * Also because of the length generalization ability of RIR-EBT-GRC, we may not need to train on long-range data. We can train on much shorter data and allow it to length generalize during inference. In fact, RIR-EBT-GRC learns even better from shorter data. RIR-EBT-GRC for example can already get near $61$% on LRA ListOps (~$2000$ sequence lengths) by training on shorter ListOps data ($\leq 100$ lengths) and possibly could get even better if LRA did not have a different number of arguments (most models still struggle to generalize to higher number of arguments. Pure length generalization performance of RIR-EBT-GRC is much better - for example, it retains $97.1$% accuracy when the only difference is sequence length such as 900-1000 sequence length split after training on data $\leq 100$ lengths).
> > >
> > > -------------
> > >
> > > Also, the point about complexity can be a bit subjective if we are not talking about any objective measure like Lines of Codes or Time/Space complexity. It is not so clear why RIR-EBT-GRC should be counted as more complicated than OM/CRvNN or popularly adopted methods in the literature like chart-based RvNNs/CYK-based RvNN [1] which utilizes dynamic programming with nested loops requiring careful implementations for efficiency or its newer extensions with beam search [2] and pruning strategies [3]. Also note that RIR-EBT-GRC, as of now, runs nearly as well without S4D. Compared to RIR-EBT-GRC w/o S4D, S4-based models or successful linear RNNs models [4,5] generally also require sophisticated initialization schemes or hybridization with Transformers with chunking [6], Flash Attention, and other modifiers like adaptive sparsification [7] for memory-efficient performant implementations.
> > >
> > > [1] Jointly learning sentence embeddings and syntax with unsupervised Tree-LSTMs - Maillard et al. Natural Language Engineering 2019
> > >
> > > [2] Unsupervised Parsing with S-DIORA: Single Tree Encoding for Deep Inside-Outside Recursive Autoencoders - Drozdov et al. EMNLP 2020
> > >
> > > [3] R2D2: Recursive Transformer based on Differentiable Tree for Interpretable Hierarchical Language Modeling - Hu et al. ACL 2021
> > >
> > > [4] Resurrecting Recurrent Neural Networks for Long Sequences - Orvieto et al. ArXiv 2023
> > >
> > > [5] How to Train Your HiPPO: State Space Models with Generalized Orthogonal Basis Projections - Gu et al. ICLR 2023
> > >
> > > [6] Mega: Moving Average Equipped Gated Attention - Ma et al. ICLR 2023
> > >
> > > [7] Sparse Modular Activation for Efficient Sequence Modeling - Ren et al. ArXiv 2023

---

### Official Review · Reviewer_jzGF · 2023-07-19

**Soundness:** 3 good
**Presentation:** 4 excellent
**Contribution:** 2 fair
**Rating:** 6
**Confidence:** 3

**Summary:**

The paper proposes a novel framework - Recursion in Recursion (RIR) for tree recursive neural networks (Tree-RvNNs) so as to get around the issue of computational infeasibility of typical RvNN models (Beam tree RvNNs are $\mathcal{O}(n)$) while still being able to exhibit length generalization on simple arithmetic tasks like ListOps. The models based on RIR have recursive depth bounded by $k\log_k n$ and still demonstrates over $90\%$ length generalization performance. They explain their method thoroughly and also compare on ListOps and LRA with a fairly large suite of baselines showing that their method perfoms comparably with the best.

**Strengths:**

1. The paper is very well written.
    - The method and related work is explained thoroughly while providing sufficient intuition.
    - If anything, I would urge the authors to explain the results a little more thoroughly even if it were at the cost of pushing some of the other content to the appendix.
2. The experiments are thorough.
    - The authors compare with a sufficiently large suite of baselines.
3. Length generalization is an important and difficult problem that existing methods (such as Transformers) struggle with. Getting RNN length generalization performance without the associated cost blowup is very interestin for the field.
4.  The authors adequately list the limitations of their work, which I appreciate.

**Weaknesses:**

1. The authors discuss the choice of $k$ somewhat briefly explaining that small $k$ surely hurts the performance of the resulting model.
    - On a few examples they try to suggest that $k=\mathcal{O}(\log n)$, however, without seeing this varied over several values of $n$, it is hard to justify this claim.
    - Note that if $k=\mathcal{O}(n)$, the method loses it's benefits.
2. The authors build on unpublished work which is cited as "Anonymous" and shared in the appendix. This is highly unusual and I have some concerns about this.
    - While there is nothing inherently wrong with this, it puts the burden of evaluating some of the claims within the paper to yet another unverified paper.
3. Some of the notation in the paper can be improved. Such as the example used for explanation in Sections 2 and 3: "$7 + 8 \times 5 - 2$". I feel this would be better if explained symbolically like $a_1 \cdot op_1\cdot a_2 \dots$"
4. Some claims are not justified very well (such as the Problem with th String-it solution on page 6)
5. I feel the results need more careful explanation. Table 1 seems like a very important result to justify the proposed method however, I don't understand why the table is split into two. (ListOps competitive and ?) Can you explain this?
6. In table 4, the proposed method RIR-GRC performs quite poorly, however this is not called out or explained.
7. Overall, the results are not particularly impressive. And while MEGA is mentioned multiple times, it does not appear to be listed in the benchmarks.
8. Efficiency and accuracy are compared in separate tables. Since the main contribution of the paper is an efficient implementation of a model that shows length generalization, I would like to see a computation vs performance tradeoff curve. I feel this would go a long way in proving the superiority of the proposed method.
9. This is very minor but I would recommend the authors include a short explanation of ListOps in the main paper. I understand that the authors chose to push it to the appendix due to space limitations but since it is such an essential part of the paper, I would recommend adding a few lines about it.

**Questions:**

1. What is "infinite receptive field" mentioned on line 37? Is there a reference for this?
2. Have you checked the scaling of $k$ with $n$? Does setting $k=\mathcal{O}(\log n)$ for some reasonably small constant consistently work?
3. Can you explain the problem with the string-it solution more clearly? I still don't follow it. Further, it seems like the beam-alignment solution applies it anyway after throwing some randomness into the mix.
4. The statement "set the chunk size according to one's computation need" seems quite vague. Can you explain?
5. What is the take away from Table 2? RIR-EBT performs well, but not as good as EBT-GRC. Is there a precise compute vs acc tradeoff here?

**Limitations:**

Yes. I commend the authors for including an honest and thorough limitations section.

---

> ### Author Rebuttal · Authors · 2023-08-10
>
> Thank you for the review. We will take the formatting suggestions into account.
>
> **Re Weaknesses:**
>
> 1. $k$ is a constant. It is the chunk size hyperparameter and is not made to depend on sequence length or the input. Yes, if we set $k$ to be very large (e.g., as large as the maximum input sequence size) then the RIR framework loses its benefit but in the paper we show that a nice trade-off can be achieved by using smaller k. Also, please see the response to Reviewer 8Jr4  for more information on $k$.
>
> 2. The cited anonymous work is published in ICML 2023. We provided an anonymous copy because the latest version of the published work was not public at the moment of submission.
>
> 4. We have added more intuition about beam alignment in the general response (+ diagrams in the general response pdf). Please let us know if you have other questions.
>
> 5. One of our main aims is to balance scalability while preserving competence in length generalization in structure-sensitive tasks like ListOps/Logical Inference. So, our target for efficiency comparison is mainly other ListOps-competent models. We show the computational cost with other methods like S4D, and Binary Balanced Tree  (which are not competent on listops length generalization or logical inference) for completeness but we put them in a different section because they are not the baselines we are most concerned with in terms of computational costs.
>
> 6. RIR-GRC is a simpler model serving as an ablation of our main RIR-EBT-GRC  proposal. It also performs worse in Tables 2 and 3 for worse structural bias. Similarly, GRC (RecurrentGRC) tends to perform worse than EBT-GRC/BT-GRC outside the RIR framework as well (results with Recurrent GRC can be found in the appendix paper “Beam Tree Recursive Cells”).
>
> * (7.1) RIR-EBT-GRC is the only model that can be feasibly run in 2000+ sequence length data and can also get 90%+ accuracy in OOD length generalization settings in ListOps (Table 2) and Logical Inference (Table 3). In contrast, S4D/MEGA gets performance within 15-30% in ListOps OOD length generalization settings (900-1000 seq length) and ~60% in logical inference. That is at least a 30% performance gap. Moreover, RIR-EBT-GRC reduces training speed and memory by approximately 30 times in 1500-2000 sequence length settings (Table 1) compared to EBT-GRC. Yes, our accuracy results in LRA IID settings are not SOTA, but the SOTA performance comes at the cost of OOD generalization at the moment.
>
> * (7.2) For some comparison with MEGA please see the general response pdf.
>
> 8. Thank you for the suggestion. We added Pareto frontier graphs in the general response pdf.
>
> **Re Questions:**
>
> 1. We will rephrase the “infinite receptive field” mentioned. We used this to refer to the unbounded nature of attention (as opposed to local convolution where interaction in a specific layer is limited to tokens within a window of some limited size).
>
> * (2-3.) We do not set $k$ as dependent on $n$. We simply choose $k$ as big as we can while getting feasible/reasonable computational costs in LRA ListOps. We set  $k=30$. Ablations with smaller $k$ are given in Table 1 Appendix. As expected, in Table 1 Appendix smaller k/chunk-size (k=20,k=10) leads to worse results but can be still better than S4D, Balanced Tree, etc. in length generalization (also EBT-GRC is effectively the result of k=infinite, and BBT-GRC is basically k=2 - and the pattern still holds that higher k yields better results at least in structure-sensitive contexts).
>
> 4. In Table 2 in the main paper RIR-EBT-GRC performs better. In Appendix Table 2, it’s a more general problem that RvNN-based models often turn out to be mostly similar in performance with plain RNNs and other alternatives when it comes to more natural language tasks (see, for example, the “Beam Tree Recursive Cells” paper in Appendix). This can indicate that RvNNs are failing to exploit syntactic structures as well in those tasks (perhaps may require pre-training with language modeling) or that these tasks have exploitable statistical shortcuts that allow simpler methods to get ahead. Still, we can see in Appendix Table 3, that in some stress tests of larger datasets like MNLI, RvNN-based models perform better than simpler GRC/RIR-GRC. So there are specific contexts where inductive biases of RvNNs tend to shine more.

---

> > ### Comment · Reviewer_jzGF · 2023-08-12
> > **Thank you for the clarifications**
> >
> > I thank the authors for their detailed rebuttal and for answering all of my questions. I am raising the score to 6 and recommending acceptance of the paper.

---

### Official Review · Reviewer_UxTS · 2023-07-22

**Soundness:** 3 good
**Presentation:** 3 good
**Contribution:** 3 good
**Rating:** 6
**Confidence:** 2

**Summary:**

The paper proposes Recursion in Recursion (RIR) to address the shortcomings that computationally efficient models like Binary Balanced Tree Recursive neural networks have in solving arithmetic tasks like ListOps. RIR also seeks to alleviate the computational burden brought forth by structure-aware ListOps-competent models such as the Beam tree RvNN. The approach, relying on 2 levels of computation, break down a sequence into chunks (of length k) on which an inner loop uses another network (e.g. Beam tree RvNN) to compute local representations. The representations from the inner loop are subsequently taken in by an outer loop. The authors suggest that this approach can help scale structure-aware inference from (n) to (k*log_k(n)), where n is the input length. The authors conducted experiments to show that their RIR-based models use much less time and memory for the ListOps task than baseline approaches. The results on ListOps, logical inference and long range arena (LRA) show that even with this efficiency, RIR’s performance is competitive with current baselines.

**Strengths:**

The paper proposes a scaleable solution that can be structure-aware and effective at tasks like ListOps.
Experiments have shown that this approach (RIR) is promising, with performance competitive with less-efficient baselines.
Overall, the paper is well-written and the contribution is clear.


**Weaknesses:**

Modifications are required to integrate existing models into the RIR framework, e.g. the EBT-RvNN, which can limit the widespread use of this approach.

**Questions:**

Typo: line 102: gaurantee -> guarantee

**Limitations:**

Yes, addressed.

---

> ### Author Rebuttal · Authors · 2023-08-10
>
> Thank you for the review.
>
> Please note the general response for clarification on the complexity of the approach.
>
> We will fix the typo.

---

> ### Comment · Reviewer_UxTS · 2023-08-12
> **Acknowledgement of rebuttal**
>
> I have read the authors' rebuttal and decided to keep my original score.

---

### Official Review · Reviewer_ozsz · 2023-07-25

**Soundness:** 2 fair
**Presentation:** 2 fair
**Contribution:** 2 fair
**Rating:** 5
**Confidence:** 4

**Summary:**

In this work, the authors try to combine the best of two worlds in proposing Recursion in Recursion, where outer recursion is K-array balanced binary tree and inner implements its cell function for recursive neural networks for sequential inputs. The proposed framework is tested on various logical inference and NLP tasks to show model can reach respectable performance and scale to longer sequences.

**Strengths:**

1. Good paper on scaling.
2. Paper is well written
3. Good sets of experiments.

**Weaknesses:**

1. Novelty is limited. The paper is more focused on scaling and several works are already published in the literature
2. Why inner cell uses BT-RvNNs, given they are similar to CYK-based RvNNs, and what advantage it offers should be mentioned in depth. Since selecting top-k is still heuristic and based on beam size, results will vary.
3. Huge standard deviation, indicating potential instability in the model.

**Questions:**

I would like to see the computational complexity analysis, such as number of FLOPs, average time for convergence for the proposed model, It is not clear what advantage proposed framework offers and what is the loss in performance. Given current paper is mainly focused on improving the scaling capability more focus should be given these problems

Effect of computation budget, can authors also report values when computational budget is higher, it is not clear how higher budget would lead to better gain, and what is the bound for it (k)
How did you choose beam size 5 and 7? Beam search should be conducted on dev set and not on test set, so how did authors come to this conclusion is challenging to understand, can authors provide more details, how did you obtain this beam search number?

The variance for RIR based model is big, it shows they are highly unstable, can authors comment on that?

In appendix authors have written “At any iteration t, we start only with some sequence”, what is some sequence, such terms is used throughout the paper, and its difficult to know what is the bound. Please provide bound or some numbers to back these claims

Why there is a need of using 2-layer MLP for fix-1 (line 133 in appendix), why not simple linear transformation, I am confused how mathematically it suffix the same function and how does it approximate the prior function?

Does beam alignment find good structures that support required compositionality? More analysis is needed on the distribution and Importance of beam alignment.

Missing relevant work [1] and [2]


1.	Mali, A., Ororbia, A.G., Kifer, D. and Giles, C.L., 2021, May. Recognizing and verifying mathematical equations using multiplicative differential neural units. In Proceedings of the AAAI Conference on Artificial Intelligence (Vol. 35, No. 6, pp. 5006-5015).
2.	Arabshahi, F., Lu, Z., Singh, S. and Anandkumar, A., 2019. Memory augmented recursive neural networks. arXiv.

**Limitations:**

1. Language can be improved, some sequence, some score, some function such terms should be avoided while explaining any mathematical concepts.
2. The variance of the proposed model is big leading to instability, hence what benefits it offers is questionable. What about the generalization effect, how does the model work on longer sentences? What is the limit? These questions are still unclear.
3. More analysis will benefit the work.

---

> ### Author Rebuttal · Authors · 2023-08-10
>
> Thank you for the review.
>
> **Re Weaknesses:**
>
> 1. Our goal is not pure scalability. Our goals are (i) to extend RvNNs in a novel manner that achieves a balance between scalability and length-generalization capacities in structure-sensitive contexts (ii) to show that it has competitive results (if not SOTA)  in LRA compared to SSM/linear RNN and generally much better results than non-hybrid Transformers.  We are unaware of other works attempting to balance scalability with length generalizability in structure-sensitive contexts. Besides, penalizing the lack of novelty of the research goal would lead to penalizing any novel method trying to improve on any pre-established research goals.
>
> 2. CYK models are much more expensive than BT-RvNN and show less performance in the non-RIR setup. This is shown in the appendix paper “Beam Tree Recursive Cells” (please see  Tables 1, 4, 5, 8 of that paper). Moreover, practical implementations of CYK-based RvNNs involve soft-attention-based marginalization of paths at every inner loop [1] or similar top-k-based selection [2] - bringing back heuristics.
>
> 3. Huge std (+-9) is only shown by a specific RIR-variant  “RIR-EBT-GRC - S4D” in a specific task - ListOpsMix (if you have some other result in mind where there is a huge std, please point it out). It’s not a general norm. Besides, even the “worst” run of that model in that task is $61.45$ which is still better than all models in Table 4 except RIR-EBT-GRC.
>
> **Re Questions:**
>
> 1. See the general response for complexity analysis. Accuracy comparisons between the models are charted for various datasets - ListOps (Table 2), Logical Inference (Table 3), and many other NLP tasks in the Appendix. Empirical time/memory comparisons between the models are charted in Table 1. Combined they show the performance-cost tradeoff. We also add Pareto frontiers in the General Response PDF.  We can add FLOPs/convergence in the final version.
>
> 2. As far as we are aware, we do not claim that a higher computational budget by itself leads to better performance. But higher chunk size ($k$) can lead to better gain and a higher computational budget can allow us to use bigger $k$. Theoretically, this is because higher $k$ leads to less of an artificial restriction from the outer balanced-tree structure (for example in the case of $k=$infinite, we recover EBT-GRC which still performs better than RIR-EBT-GRC in structure-sensitive tasks at the cost of memory and time). Empirically, we show this in ablation Table 1 (where reducing chunk size $k$ to 20/10/2 from 30 leads to worse performance in general (k=2 is equivalent to BBT-GRC)).
>
> 3. A higher beam size can expand the search space and enhance error recovery (prevent chances of filling the beams with all bad structures). As such, higher was chosen a priori because RIR makes it more feasible to use a higher beam size.
>
> 4. We found that “RIR-EBT-GRC - S4D” fails in utilizing the low-sequence-length data as well (but still better than S4D and others) in ListOpsMix in some runs but utilizes it much better in others. Otherwise, in general, we do not observe high standard deviation for RIR models in general or even “RIR-EBT-GRC - S4D” in general (see other LRA experiments for example besides ListOpsMix). Also, please refer to point 3 in Re Weaknesses.
>
> 5. “Some sequence” refers to a sequence $\in \mathbb{R}^{n \times d}$ (n being the sequence size). We will add more clarifying notations.
>
> 6. Linear transformation can be used but we haven’t tried it. We used a non-linear function because linear functions are less expressive and because the original scoring also involved non-linearity. We do not have to “approximate” the original function in any strict sense - we just need a reasonable scoring function that gets the same input format and can be trained with gradient descent. The point is that the original function is computationally more expensive and without any clear a priori motivation for using it - and at the same time, empirically (i.e., a posteriori) it doesn’t show much better accuracy (as shown in various EBT-GRC vs BT-GRC experiments in the main paper and the appendix).
>
> 7. Strictly speaking “good structures” in a human interpretable sense is abandoned in RIR-framework (unless one uses non-RIR inference but then beam alignment will not be relevant during inference) because of enforced balanced tree outer recursion. Beam alignment simply allows alignments of beams  from different chunks to be sensitive to beam score. The usefulness of beam alignment is empirically shown in Ablation Table 1 (in appendix). Both random alignment (+Random Align) and removal of beam alignment (-Beam Align) result in poorer performance. More intuition (with diagrams in the general response pdf) on beam alignment is provided in the general response.
>
> We will add the citations you mentioned. Thank you.
>
> [1] Jointly Learning Sentence Embeddings and Syntax with Unsupervised Tree-LSTMs - Maillard et al., Natural Language Engineering 2019
>
> [2] Unsupervised Parsing with S-DIORA: Single Tree Encoding for Deep Inside-Outside Recursive Autoencoders - Drozdov et al. EMNLP 2020

---

### Official Review · Reviewer_619V · 2023-07-27

**Soundness:** 3 good
**Presentation:** 3 good
**Contribution:** 3 good
**Rating:** 5
**Confidence:** 4

**Summary:**

This paper addresses the long-sequence modeling problem. A recursion in recursion strategy is proposed to balance the advantages between BB-Tree RvNNs and RvNN models. The idea is straightforward but achieves competitive performance on LRA tasks.

**Strengths:**

1. The paper is easy to follow.

**Weaknesses:**

1. There are large numbers of related works missing. For long sequence modeling, there are at least 4 types of methods, including Linear attention, SSM, Linear RNN, and LongConv. Since these methods are implemented for the same goal, they should be included in related work as well as the experiment section.
2. The experiments are inadequate. LRA is a toy benchmark for assessing long-sequence modeling. It is insufficient to use it as the sole indicator of effectiveness. In fact, this work focuses solely on a sub-task in LRA, making the experiments even weaker. I would encourage the author to verify the actual long sequence modeling capabilities in real-world scenarios such as language modeling, image classification, etc.
3. Linear RNN has achieved STOA performance in many benchmarks, including LRA, language modeling, and image classification, as demonstrated in recent papers. Why should we consider non-linear RNNs, which are slow to train and perform no better than linear RNNs?
4. The concept is straightforward and straightforward. By forming a RIR structure, it combines the advantages of the two methods. As a result, the processing time is lengthened. It would be more appealing if the processing time could be shortened. Furthermore, the competitive methods are ineffective. I don't see any standard benchmark LRA results, so it's difficult for me to justify the effectiveness of the proposed method.
5.The maximum sequence length used in this paper for an efficient long sequence modeling method is 2K, which is a standard sequence length for transformer LLM. It would be preferable to see the proposed method in long sequence tests, such as 32K and higher. Furthermore, long sequence modeling takes much longer to process than transformer, making the method less appealing in real-world scenarios.

**Questions:**

See the weakness part.

**Limitations:**

Yes.

---

> ### Author Rebuttal · Authors · 2023-08-10
>
> Thank you for the review.
>
> * (1.1) Many of the models you mention are already cited. SSM models are cited in [15,16,17,40], LongConv models are cited in [39,12], Linear RNN is cited in [34,30]. Top efficient Transformer based models are cited in [49,9] + indirectly efficient Transformers are referred through [45].
>
> * (1.2) We will add more citations and discussion in related work. We will add more models in LRA.
>
> * (1.3) Our main goal is balancing scalability with the ability for robust length generalization in structure-sensitive tasks. Correct us if we are wrong, but to our knowledge, this is not tackled by any of the SSM, Linear RNN, LongConv, or Efficient Transformer models. We compare S4D as a representative of LinearRNN/SSM models for length generalization settings, and we also added MEGA (see the general response pdf). Prior results [p5,p6] have already shown that full Transformer models (that efficient transformers try to approximate) don’t work for length generalization in structure-sensitive contexts.
>
> * (2.1) Text and Retrieval tasks in LRA are both realistic tasks - one is IMDB sentiment classification and another AAN document retrieval. Neither is obviously any more toyish than image classification tasks like Sequential Cifar. Besides, “toy tasks” can be important tasks when most existing models struggle with them or fail to learn them in a generalizable manner (eg. logical inference or ListOps) without being inefficient (like Ordered Memory, CRvNN, or original Beam Tree Cell). Besides, in the Appendix we also show experiments on Natural Language Inference, Semantic Similarity, and other sentiment classification datasets. Currently, RIR-EBT-GRC is a sentence encoder model. Scaling it for causal language modeling would require more modifications which would be an interesting direction to explore  for future work.
>
> * (2.2)  It is worth noting that modern RvNNs (CRvNN, BT-RvNN) are relatively less mature. Just because they have some limitations now - doesn’t mean we will not find workarounds later. Please note that even predecessors of modern SSM-based models/S5 [p1,p2,p3] showed limited applicability or limited empirical demonstrations before being picked up and enhanced in the future. Our modeling approach is less mature and would require more development. What we show is that it has a potential in length generalization that is not shown by others (see point 3).
>
> * (3.) “Why should we consider non-linear RNNs, which are slow to train and perform no better than linear RNNs?” - As we demonstrate: our non-linear RvNNs such as RIR-EBT-GRC perform better than Linear RNNs (S4D) in OOD settings or in learning from shorter length data in Table 2 (ListOps)/Table 4 (ListOpsMix) and Table 3 (Logical Inference). We also show that RIR-EBT-GRC performs better than MEGA in similar settings (please see general response pdf).
>
> * (4.1) Regarding RIR complexity, see the general response. As expected from the theoretical time complexity, RIR-X speeds up any recursive model X by an order of magnitude (please see Table 1 in the main paper) - and thus reduces processing time.
>
> * (4.2) You are right that other ListOps-competitive RvNNs are inefficient for LRA. But that’s precisely what justifies the efficiency of RIR-X models in comparison to prior RvNNs. Besides that, we have also shown comparisons between RIR-X and X models in Table 1, Table 2, and other natural language tasks in the appendix - all of which show where our main RIR-based models stand in contrast to prior RvNNs in tasks where we can compare them. In the general response pdf, we also added Pareto frontiers showing the effectiveness of RIR-X models.
>
> * (5.) The maximum length for IMDB (LRA Text)  is 4000 (not 2K). Our model will not be as efficient with 32K+ but please see point 2.2 above on this issue. Moreover, efficiency seems to come at other costs like OOD robustness (point 3).  One future extension for RIR-framework could be to compress the sequence length beforehand like Funnel Transformer [p4] before running the model to make it handle larger sequence lengths.
>
> [p1] HiPPO: Recurrent Memory with Optimal Polynomial Projections - Gu et al. Neurips 2019
>
> [p2] Legendre Memory Units: Continuous-Time Representation in Recurrent Neural Networks -Voelker  Neurips 2019
> [p3] Parallelizing Linear Recurrent Neural Nets Over Sequence Length - Martin et al. ICLR 2018
>
> [p4] Funnel-Transformer: Filtering out Sequential Redundancy for Efficient Language Processing - Dai et al. Neurips 2020
>
> [p5] The Importance of Being Recurrent for Modeling Hierarchical Structure - Tran et al. EMNLP 2018
>
> [p6] Ordered Memory - Shen et al. Neurips 2019

---

> > ### Comment · Reviewer_619V · 2023-08-19
> > **Official Comment by Reviewer 619V**
> >
> > Thanks to the author for the reply, my concerns have been addressed and I will raise the score to 5

---

### Official Review · Reviewer_8Jr4 · 2023-07-27

**Soundness:** 3 good
**Presentation:** 4 excellent
**Contribution:** 3 good
**Rating:** 7
**Confidence:** 4

**Summary:**

This paper introduces the recursion-in-recursion (RIR) framework for balancing the tradeoffs between
1. sequential processing, which offers better inductive bias and stronger solutions for many types of symbolic processing and logical inference tasks, but can be very expensive
2. balanced tree recursion, which shortens the length of the computational graph and can be fast and scalable, but struggles with the aforementioned tasks

The RIR framework is thoroughly evaluated on a simple set of arithmetic tasks (ListOps), where it shows the ability to solve and generalize on this task, while being much faster computationally than previous methods with this ability.


**Strengths:**

- The paper provides a very well-explained exposition of a lesser-known model approach.
- The proposed method/framework is novel and makes intuitive sense.
- The method performs very well on symbolic processing tasks that it was designed for. It almost preserves the performance of much more expensive methods (e.g. full beam search) while being an order of magnitude faster.
- Results on other LRA tasks involving language are also shown, indicating that the method is not necessarily specialized to synthetic symbolic processing tasks but could be a viable more general approach


**Weaknesses:**

A weakness of the method itself is that the RIR framework adds complexity because of the two separate levels of hierarchy which can be freely chosen. Additionally the $k$ hyperparameter seems very important and there doesn't seem to be a first-principles way to choose it well. It seems like even if there is a model that can perfectly solve a given task, once RIR is introduced there are no guarantees about whether the task can be perfectly solved. Thus it becomes a heuristic tradeoff between efficiency and strength, with many hyperparameters that must be managed.


**Questions:**

I think it help the story to have some attention baselines. Although many variants of attention/transformers have been tried in the original LRA works, showing some ablations within the RIR framework (e.g. as either the inner or outer aggregator) seems interesting. This is interesting particularly because attention is often viewed as a catch-all solution to discrete and symbolic data. However I am unlikely to increase my score even if these are shown, and this is just a suggestion that could strengthen the paper and make the overall line of work more convincing.


**Limitations:**

Main limitations are properly addressed. It might be worth being more explicit about the fact that the proposed family of methods is not meant to address other types of sequential data such as perceptual signals (e.g. images/audio) and likely doesn't work in those settings.  (Otherwise people may also wonder why the particular subset of LRA was chosen.)

---

> ### Author Rebuttal · Authors · 2023-08-10
>
> Thank you for the review.
>
> * Please check the general response for comments on the complexity of RIR.
>
>
> * Regarding the selection of the hyperparameter $k$, a higher $k$ (chunk size) should be generally better in structure-sensitive tasks. $k=$infinite (or maximum input sequence length) for example would reduce RIR-EBT-GRC to EBT-GRC (which is still better in ListOps). A lower $k$ leads to lower accuracy but higher computational performance. So one first-principle way to choose $k$ is to just settle on the maximum value whose computational performance one is comfortable with. In Appendix Table 1 Ablation we show that reducing $k$ leads to lower performance, as expected, on ListOps  (please see chunk size = 20 or 10).
>
>
> * RIR + Transformer is an interesting area to explore more extensively that we keep for future work. There is a contemporary work that independently came up with a similar idea that they use to achieve OOD generalization on parity tasks [1] - this shows the general potential for the RIR framework. We briefly tried RIR + Transformer once in ListOps but it wasn’t as promising.
>
>
> * We were mainly focusing on language processing tasks where hierarchical bias is relevant (it can be relevant in image domains as well, but it’s more tricky in LRA since the images are flattened). We believe some more work is needed for RvNNs to work well there. Nevertheless, we ran them on CIFAR from LRA, and RvNN-based models can get 60%+ which is worse than the SSM-based/LongConv models but much better than any non-hybrid Transformer models. We will add some more analyses and discussion regarding that.
>
> [1] Transformer Working Memory Enables Regular Language Reasoning And Natural Language Length Extrapolation - Chi et al. ArXiv 2023

---

> > ### Comment · Reviewer_8Jr4 · 2023-08-21
> >
> > Thanks for the rebuttal. I am maintaining my score on the grounds that although perhaps not immediately practical, this work explores an important direction toward structure-sensitive sequence modeling problems that is not sufficiently addressed by existing methods.

---

### Author Rebuttal · Authors · 2023-08-10

We thank all our reviewers for their insightful reviews and comments.

**On the complicacy of RIR:**

* **RIR Computational Complexity:** RIR-X is generally computationally more efficient than X both memory-wise and time-wise. For example, this can be observed in RIR-EBT-GRC vs EBT-GRC in Table 1. If the time complexity of X is $O(f(n))$ (n being the sequence size), the time complexity for RIR-X is  $O(\log_k{n} f(k))$. This can effectively turn into $O(\log{n})$ if $k$ is a constant.

* **RIR Implementation Complexity:** RIR is only slightly more complicated to implement. Particularly, balanced tree recursion is a simple algorithm, and all we have to do is import some existing code for inner recursion within the balanced tree recursion for the 2-level recursion. In the case of BT-RvNN, there is an extra complexity for beam alignment but that too is only a few lines of code. Adding pre-chunk S4D contextualization is optional and requires adding only 2-3 lines of code if we are importing from an existing implementation of the inner recursion model.

* Overall, implementation of RIR-EBT-GRC (without S4D) from scratch can still be simpler (or as complex) as prior approaches like Ordered Memory or CRvNN. Also, the code is shared in Supplementary Materials (and will be open-sourced in GitHub with documentation) to allay implementation difficulties.

**Updates in the Rebuttal PDF (attached below)**:

1. We added MEGA results on ListOps (Table 1)  and Logical Inference (Table 2). As can be seen from these results, MEGA performs better than S4D but still falls behind RIR-EBT-GRC by a large margin in structure-sensitive length generalization contexts.

2. We added scatter plots + Pareto frontiers in Fig. 2. In all three subfigures (focusing on different trade-offs)  in Fig. 2, RIR-EBT-GRC (RRE in the figures) remains as a Pareto-efficient solution that maintains a highly competitive trade-off. S4D, BBT-GRC, and RIR-GRC can win on the time cost and memory cost compared to RIR-EBT-GRC, but with a sharp degradation of OOD performance in structure-sensitive tasks (logical inference, ListOps). While OM, CRvNN, BT-GRC, BT-GRC OS, and EBT-GRC can outperform RIR-EBT-GRC (to an extent) on OOD length generalization accuracy in ListOps and logical inference, they come with much more exorbitant time/memory cost.
We also added more clarification diagrams for beam alignment in Figures 1a, 1b. That is, in Fig 1a, we show the visualization of the String-it solution and in Fig 1b, we show the visualization of the beam alignment solution.

**Additional Beam Alignment Intuitions:**

RIR framework allows a degree of parallel processing - multiple non-overlapping chunks can be simultaneously processed by the inner recursion cell.

However, when the inner recursion cell is EBT-GRC, each chunk will return different lists of $b$ beams (sorted in the descending order of their scores). If there are $m$ chunks, we will have $m$ lists where each list will have $b$ beams. Each beam should have a scalar beam score and a beam sequence $\in \mathbb{R}^{1 \times d}$ where $1$ is the sequence size and $d$ is the hidden state size. But for the next iteration, we will need a single list of $b$ beams (where each beam should have a scalar beam score and a beam sequence $\in \mathbb{R}^{m \times d}$ where $m$ is the sequence size and $d$ is the hidden state size. Note that $m$ is initially the number of chunks. After combining the outputs of the chunks, we get the sequence size. Since output of each chunk has sequence size 1, concatenating output of m chunks lead to m sequence size.). The question then is - how to combine the different $m$ lists of $b$ beams into a single list of $b$ beams.

In the visualizations, we consider a simple scenario where $m=2, b=3$. The String-it solution (Fig. 1a), is to simply concatenate the beam sequences (and add the corresponding scores) from all lists that occur in the same position. For example, in Fig. 1a we concatenate the first beam (Beam Sequence 1) from the first list (chunk 1 results) with the first beam (Beam Sequence 4) from the second list (chunk 2 results). Similarly, we concatenate the second beam from the first list with the second beam from the second list, and so on.

However, we want to create "high-scoring" beam combinations with more probability. However, the string-it solution does not care for that. For example, ``Beam Sequence 1 + Beam Sequence 5" would be the second highest-scoring beam, but it cannot be ever selected by string-it since Beam Sequence 1 is in the first position of the first list, and Beam Sequence 5 is in the second position of the second list - that is, they are not in the same position.

Thus, we propose Beam Alignment where we take a stochastic approach towards the ideal of biasing the preservation of high-scoring combinations. For this approach, instead of immediately applying the string-it solution, we make the beams in each list stochastically 'compete' for their positions. Essentially, we want high-scoring beams to be more likely to occupy more positions in the beam lists from each chunk. This is done by simply sampling a beam per list position (for each of the $b$ positions) according to the normalized beam score distribution. The result is that the beam lists will be now filled with mostly the higher-scoring beams (See the results after `Sample' in Figure 1b). Next, if we simply apply the string-it solution at this point, it automatically leads to high-scoring combinations as we wanted because of the prior sampling of high-scoring combinations. Now there is a possibility of combinations like ``beam sequence 1 + beam sequence 5" to arise because the sampling step allows the same beam to occupy multiple positions.

Overall as we can see, in the simulation of Beam-alignment (Fig. 1b) the resulting combined beams tend to have higher scores than in the case of the direct application of String-it (Fig. 1a).

---

### Decision · Program_Chairs · 2023-09-21

**Decision:**

Accept (poster)

**Comment:**

This paper received many reviews and generally there have been good evidence and support for the acceptance of this paper.